# The ubiquitylation of IL-1β limits its cleavage by caspase-1 and targets it for proteasomal degradation

Swarna L. Vijayaraj[1,2,6], Rebecca Feltham [1,2,6], Maryam Rashidi[1,2], Daniel Frank[1,2], Zhengyang Liu [1], Daniel S. Simpson[1,2], Gregor Ebert [1,2], Angelina Vince[1], Marco J. Herold[1,2], Andrew Kueh[1,2], Jaclyn S. Pearson [3,4,5], Laura F. Dagley [1,2], James M. Murphy [1,2], Andrew I. Webb [1,2], Kate E. Lawlor [1,2,3,5,7✉] & James E. Vince [1,2,7✉]

Interleukin-1β (IL-1β) is activated by inflammasome-associated caspase-1 in rare autoinflammatory conditions and in a variety of other inflammatory diseases. Therefore, IL-1β activity must be fine-tuned to enable anti-microbial responses whilst limiting collateral damage. Here, we show that precursor IL-1β is rapidly turned over by the proteasome and this correlates with its decoration by K11-linked, K63-linked and K48-linked ubiquitin chains. The ubiquitylation of IL-1β is not just a degradation signal triggered by inflammasome priming and activating stimuli, but also limits IL-1β cleavage by caspase-1. IL-1β K133 is modified by ubiquitin and forms a salt bridge with IL-1β D129. Loss of IL-1β K133 ubiquitylation, or disruption of the K133:D129 electrostatic interaction, stabilizes IL-1β. Accordingly, *Il1b^{K133R/K133R}* mice have increased levels of precursor IL-1β upon inflammasome priming and increased production of bioactive IL-1β, both in vitro and in response to LPS injection. These findings identify mechanisms that can limit IL-1β activity and safeguard against damaging inflammation.

[1] The Walter and Eliza Hall Institute of Medical Research, Parkville, VIC, Australia. [2] Department of Medical Biology, University of Melbourne, Parkville, VIC, Australia. [3] Centre for Innate Immunity and Infectious Diseases, Hudson Institute of Medical Research, Clayton, VIC, Australia. [4] Department of Microbiology, Monash University, Clayton, VIC, Australia. [5] Department of Molecular and Translational Science, Monash University, Clayton, VIC, Australia. [6] These authors contributed equally: Swarna L. Vijayaraj, Rebecca Feltham [7] These authors jointly supervised this work: Kate E. Lawlor, James E. Vince. ✉email: kate.lawlor@hudson.org.au; vince@wehi.edu.au

nterleukin-1β (IL-1β) is a myeloid-derived pro-inflammatory cytokine that coordinates innate and adaptive immune responses to clear pathogens. However, when produced in excess IL-1β drives both complex common inflammatory diseases, such as atherosclerosis, rheumatoid arthritis and gout, and monogenic inflammatory disorders, including Cryopyrin-associated periodic syndrome (CAPS)[1,2]. Evidence also suggests that IL-1β is pathologically activated in neurodegenerative conditions, such as Parkinson's and Alzheimer's diseases. This knowledge prompted the development of biologicals that directly target IL-1β which have had considerable success in the clinic[3,4]. However, alternative IL-1β-targeted therapeutics are required with improved pharmacokinetic profiles that, in particular, can solve the problem of inadequate blood brain barrier penetrance of current biologicals to treat neuroinflammation[4,5]. Understanding the cellular signals that result in IL-1β production, or regulate its stability and turnover, are therefore important for unraveling its role in health and disease, and for the design of next-generation therapeutics.

Most of our knowledge on how IL-1β is regulated stems from the study of inflammasomes. Inflammasomes are cytosolic multimeric complexes that are required for caspase-1-mediated proteolysis of inactive full-length precursor IL-1β (pro-IL-1β) to its mature bioactive (p17) form[6]. At steady state, however, cellular IL-1β expression is often undetectable and the ligation of pattern recognition receptors (PRR), such as Toll-like receptors (TLRs), is required to induce transcription of inactive pro-IL-1β. This signal, referred to as inflammasome priming, can also increase the expression of other inflammasome components, including the inflammasome sensor NOD-like receptor protein 3 (NLRP3). The activation of inflammasome sensor proteins is triggered by a second signal, which include a diverse range of microbial and host-derived danger molecules. These activating ligands promote the association of caspase-1 with an inflammasome sensor protein, via homotypic Caspase activation and recruitment domain (CARD) interactions or, as is the case for NLRP3, via the bipartite adapter, apoptosis-associated speck-like protein containing a CARD (ASC). Subsequently, Caspase-1 undergoes proximity-induced autoproteolysis and cleaves IL-1β, as well as the related IL-1 family member, IL-18, to release their bioactive fragments[7–9]. At the same time, Caspase-1 processes Gasdermin D (GSDMD), thereby allowing the N-terminal GSDMD domain to form pores in the plasma membrane, which contribute to the egress of IL-1β and also cause the lytic form of cell death known as pyroptosis[10–13]. These events—the transcriptional induction of inactive pro-IL-1β, the critical requirement of IL-1β cleavage for its activation, and the formation of GSDMD pores to promote activated IL-1β release—underscore the importance of tightly regulating cellular IL-1β to prevent pathological outcomes.

Most cytokines are controlled at the transcriptional level and, upon expression, are targeted into the secretory pathway for release. This limits their potential for regulation by cellular post-translational mechanisms, such as modification by ubiquitin or phosphorylation, which can alter protein turnover and signaling[14]. However, IL-1β is referred to as a leaderless protein, as it lacks the signal sequence that is required for packaging into the conventional secretory pathway[15]. Instead, a polybasic motif in IL-1β directs it to the plasma membrane for release via GSDMD-dependent and GSDMD-independent mechanisms that have not been fully elucidated[15,16]. The fact that cytosolic pro-IL-1β exists alongside other inactive inflammasome components within a cell raises the intriguing possibility that it may undergo post-translational modifications to regulate its activity. In support of this idea, recent studies have associated the deubiquitinase (DUB) activity of A20 with the repression of NLRP3 inflammasome function and IL-1β activation[17], and A20 haploinsufficiency or deletion can drive damaging hyperinflammation in humans

and mice[18–21]. Nevertheless, a detailed mechanistic analysis of pro-IL-1β post-translational modification is required to determine how inflammatory responses may be fine-tuned.

In this study we show that pro-IL-1β expression, its decoration by K11-linked, K63- linked and K48-linked ubiquitin chains, and its consequent proteasomal targeting, are a general feature of inflammasome priming, and that these events are further exacerbated by canonical NLRP3 inflammasome activators. Unexpectedly, we also show that ubiquitylated pools of pro-IL-1β cannot be efficiently cleaved by caspase-1 for activation. IL-1β ubiquitylation on lysine K133 promotes IL-1β turnover, and loss of IL-1β K133 ubiquitylation via the generation of $Il1b^{K133R/K133R}$ mice enhances i) pro-IL-1β protein levels, ii) the capacity of IL-1β to be activated following NLRP3 triggering, and iii) increases levels of precursor and activated IL-1β following LPS injection. Therefore, the decoration of IL-1β by ubiquitin chains is an important safeguard that limits inflammasome-driven IL-1β inflammatory responses.

## Results

**Inflammasome priming results in the ubiquitylation of precursor IL-1β and NLRP3.** TLR signaling primes inflammasome activation by inducing the cellular production of cytosolic, inactive, precursor IL-1β, and NLRP3. To examine if inflammasome priming might result in the ubiquitylation of core inflammasome machinery, bone marrow derived macrophages (BMDMs) were treated with the TLR4 ligand LPS and the ubiquitylated proteome purified using agarose-coupled Tandem Ubiquitin Binding Entities (TUBEs). As expected, LPS treatment induced the upregulation of both NLRP3 and IL-1β, while expression of the other inflammasome components, ASC and caspase-1, remained comparatively unchanged (Fig. 1a, lysate input). Within 3–6 h of LPS stimulation, both NLRP3 and IL-1β high-molecular weight laddering, most likely reflecting their ubiquitylation, accumulated in the TUBE purified ubiquitylated proteome of both primary BMDMs (Fig. 1a) and immortalized BMDMs (iBMDMs) (Fig. 1b). On the other hand, caspase-1 and ASC modification was not detected either prior to or post LPS treatment (Fig. 1a). Of note, agarose control beads (lacking conjugated ubiquitin-associated domains [UBAs]) demonstrated the specificity of this approach, as they failed to purify the ubiquitylated proteome and any inflammasome component (Fig. 1a). Importantly, a further inflammasome priming stimuli Pam₃Cys (Pam3CysSerLys4) that activates TLR1/2 signaling, also resulted in both NLRP3 and IL-1β high-molecular weight laddering (Supplementary Fig. 1a), suggesting these modifications are a general feature of inflammasome priming.

To determine if TLR engagement and resultant precursor IL-1β production induces IL-1β modification simply through the expression of IL-1β, or if other TLR signals are required, we complemented $Il1b^{-/-}$ iBMDMs with a constitutively expressed $Il1b$ plasmid and examined IL-1β modification with and without LPS stimulation. Notably, TUBE purification demonstrated that IL-1β expression alone sufficed to trigger IL-1β modification, and this was not altered by LPS stimulation (Fig. 1c), despite LPS efficiently inducing NLRP3 ubiquitylation in both IL-1β complemented and $Il1b^{-/-}$ iBMDMs (Fig. 1c).

Purification of ubiquitylated proteins via the TUBE/UBA-domain approach may indirectly co-purify ubiquitin-associated proteins that are targeted by other molecules, such as the ubiquitin modifiers NEDD8 and SUMO; which can also cause increases in the molecular weight of targeted proteins[22]. Indeed, HA-tagged NEDD8 or ubiquitin, but not SUMO2 or SUMO3, immunoprecipitated modified high-molecular weight IL-1β when expressed in 293T cells (Supplementary Fig. 1b). To evaluate if the endogenous high-molecular weight IL-1β we detected upon

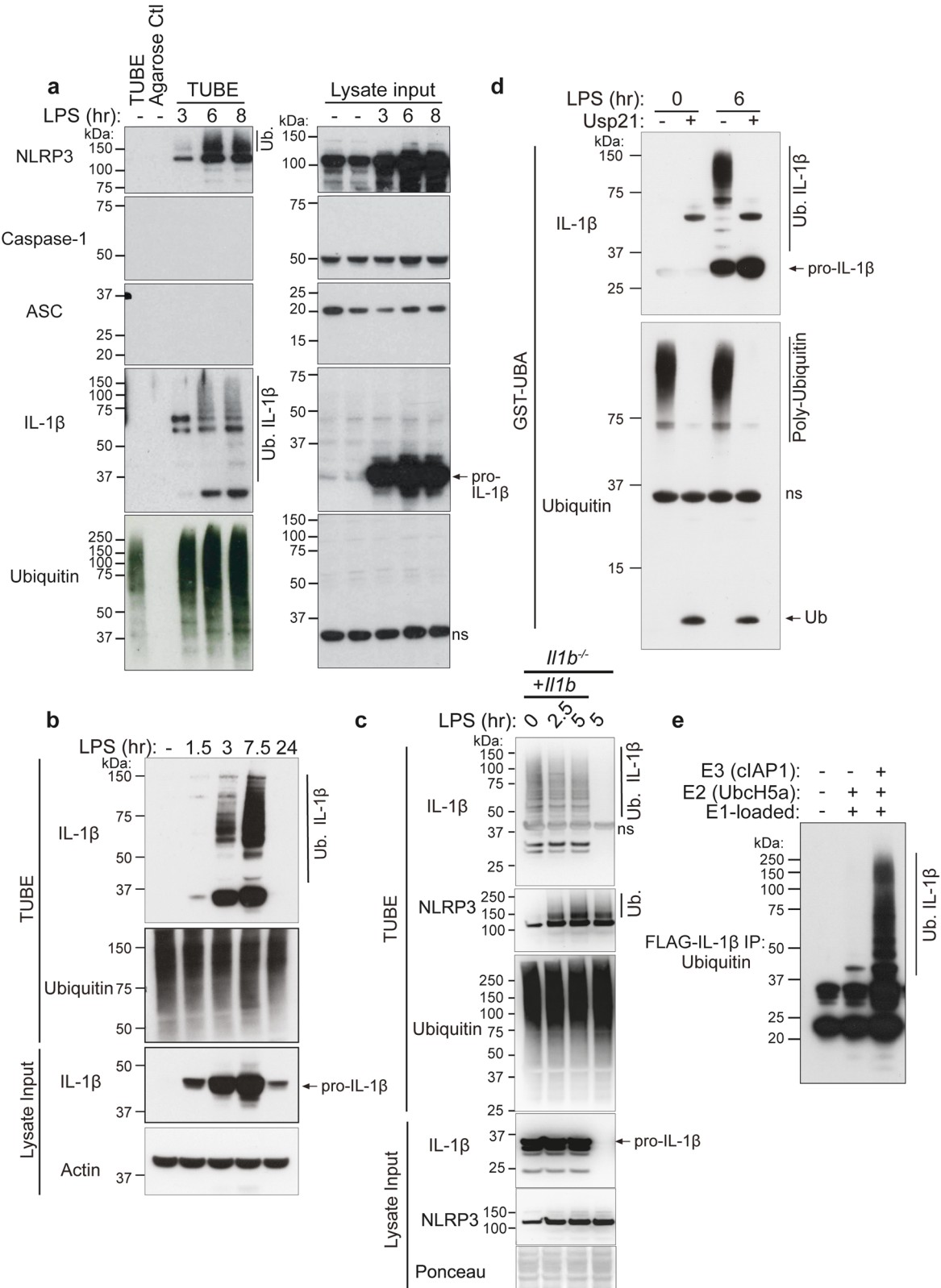

inflammasome priming of macrophages represents its ubiquitylation, we stimulated iBMDMs with LPS to induce IL-1β expression, purified the modified IL-1β using GST-UBA, and then treated samples with a ubiquitin specific peptidase, USP21, which cleaves all ubiquitin chain types (Fig. 1d). Consistent with the idea that endogenous IL-1β is targeted for ubiquitylation, rather than NEDDylation or SUMOylation, USP21 addition resulted in the complete collapse of high-molecular weight IL-1β species (Fig. 1d). Moreover, purified FLAG-tagged IL-1β incubated in vitro with recombinant ubiquitin-activating (E1), ubiquitin-carrier (E2), and ubiquitin-ligase (E3) proteins generated substantial IL-1β ubiquitylation (Fig. 1e). These results establish that upon inflammasome priming endogenous IL-1β is decorated with ubiquitin chains.

**Fig. 1 TLR-induced inflammasome priming triggers IL-1β and NLRP3 ubiquitylation. a** BMDMs were primed with Lipopolysaccharide (LPS, 50 ng/ml) for up to 8 h (hr) and ubiquitylated (Ub.) proteins were isolated from cell lysates by Tandem Ubiquitin Binding Entity (TUBE) purification. Immunoblots were performed on cell lysates (input) and purified ubiquitylated proteins (TUBEs) for the indicated proteins. Agarose control (ctl) shows the specificity of ubiquitylated protein purification. One of two experiments. **b** Immortalized BMDMs (iBMDMs) were treated with LPS (50 ng/ml) for up to 24 h and ubiquitylated proteins were purified from cell lysates using TUBE purification. Immunoblots were performed on cell lysates (input) and purified ubiquitylated proteins (TUBEs) for the indicated proteins. One of three experiments. **c** IL-1β deficient iBMDMs were infected with a retroviral *Il1b* cDNA vector containing an internal ribosome entry site (IRES) upstream of Green Florescent Protein (GFP). The complemented cells were sorted for GFP expression and stable cell lines established. Parental *Il1b*$^{-/-}$ or complemented (*Il1b*$^{-/-}$ + *Il1b*) iBMDMs were treated with LPS (100 ng/ml) for the indicated times and IL-1β and NLRP3 modification examined by TUBE purification and immunoblotting. One of two independent experiments. **d** iBMDMs were treated with LPS (50 ng/ml) for 0 or 6 h and ubiquitylated proteins immunopurified from cell lysates using a Glutathione S-transferase-Ubiquitin Associated domain (GST-UBA) fusion protein. Samples were treated with the ubiquitin specific peptidase USP21 to cleave ubiquitin from isolated proteins. Immunoblots were performed and probed for indicated proteins. One of three experiments. **e** FLAG-IL-1β was incubated, as indicated, with recombinant E1 ubiquitin activating enzyme, E2 conjugating enzyme UbcH5a and E3 ubiquitin ligase cellular IAP1 (cIAP1), and the conjugation of ubiquitin onto FLAG-IL-1β analyzed by immunoblot. One of three experiments. ns, non-specific band.

**Ubiquitylated IL-1β is rapidly degraded by the proteasome.** The ubiquitylation of proteins often marks them for degradation by targeting them to the proteasome or lysosome for proteolytic destruction. Therefore, to examine if the timing of ubiquitylation of IL-1β or NLRP3 (Fig. 1) correlated with the degradation and turnover of these inflammasome components, we primed BMDMs with LPS to induce IL-1β and NLRP3, then abrogated protein synthesis by treating cells with cycloheximide (CHX) over time. Notably, within 2–6 h of CHX treatment, total precursor IL-1β was substantially depleted from cells (Fig. 2a). On the other hand, NLRP3, ASC, and Caspase-1 levels were not impacted, while the other inflammasome-associated cytokines, IL-18 and IL-1α, were moderately reduced after 6 h of CHX treatment (Fig. 2a).

To define if IL-1β turnover was mediated by proteasomal or lysosomal targeting, BMDMs were primed with LPS to induce IL-1β expression, then treated with CHX in the absence or presence of the proteasomal inhibitor MG132, or the inhibitor of lysosomal function, Bafilomycin A1 (BafA1). Notably, only MG132-mediated proteasome blockade substantially protected IL-1β from degradation, similar to its ability to prevent the turnover of the short-lived anti-apoptotic protein Mcl-1 (Fig. 2b). In contrast, Bafilomycin A1 treatment had no impact on IL-1β turnover and did not increase IL-1β stability further when combined with MG132 treatment (Fig. 2b). Unsurprisingly, ASC and NLRP3 levels were not significantly altered by either proteasomal or lysosomal inhibition following the arrest of protein synthesis (Fig. 2b).

LPS priming of BMDMs, followed by MG132 treatment over time, augmented the detection of ubiquitylated IL-1β protein (Fig. 2c), consistent with ubiquitylation targeting IL-1β for proteasomal clearance. Typically, proteins marked with K48-linked ubiquitin chains are targeted to the proteasome, although K63-linkages and K11-linkages have also been implicated[22–24]. To determine if IL-1β undergoes K48-linked ubiquitylation and/or is modified with other ubiquitin linkage types, ubiquitylated IL-1β was isolated from macrophages stimulated with LPS and MG132 and subjected to treatment with deubiquitinating enzymes specific for K63-linked (AMSH), K48-linked (OTUB1), K11-linked (Cezanne), and M1 linear-linked (OTULIN) ubiquitin chains (Fig. 2d)[22]. Treatment with AMSH, OTUB1, and Cezanne all reduced, to varying extents, high molecular weight IL-1β ubiquitylation, whereas OTULIN treatment to remove linear ubiquitin chains had no impact. As expected, treatment of ubiquitylated IL-1β with the ubiquitin peptidase USP21, which is capable of cleaving all ubiquitin linkage types, or vOTU which recognizes all linkage types except M1 linked (or linear) ubiquitin chains, completely collapsed high molecular weight modified IL-1β (Fig. 2d). Similarly, the purification (under denaturing conditions)

of ubiquitylated proteins using antibodies that recognize only K63-linked or K48-linked ubiquitin chains also isolated endogenous, ubiquitylated, IL-1β (Supplementary Fig. 2). Therefore, LPS not only primes the inflammasome by upregulating IL-1β transcriptionally but also triggers post-translational K63-ubiquitin, K48-ubiquitin, and K11-ubiquitin chain modification of IL-1β that correlate with its proteasomal degradation at a rate comparable to other short-lived proteins, such as Mcl-1[25].

**ATP and nigericin augment Caspase-1-independent IL-1β proteasomal degradation.** The finding that LPS stimulation may facilitate ubiquitin-mediated IL-1β turnover led us to query whether NLRP3 inflammasome activators, such as ATP and nigericin, might also alter the turnover of IL-1β to decrease or increase its inflammatory potential. As one would expect, ATP or nigericin treatment of LPS-primed wildtype (WT) BMDMs resulted in the rapid loss of both cellular precursor IL-1β and IL-18 pools (Fig. 3a) due to caspase-1-mediated IL-1β and IL-18 cleavage, pyroptotic cell death, and unconventional cytokine secretion. Thus, to identify non-proteolytic regulatory mechanisms, we derived BMDMs from mice lacking inflammasome machinery, Caspase-1 (and Caspase-11) or NLRP3, which are devoid of the capacity to rapidly cleave IL-1β and IL-18 and cannot undergo pyroptotic cell death in response to canonical NLRP3 inflammasome activators. Notably, over time, ATP and nigericin treatment specifically reduced precursor IL-1β levels even in inflammasome-deficient BMDMs, and this degradation was limited by MG132-mediated inhibition of proteasome activity (Fig. 3b, c and Supplementary Fig. 3a). In contrast to enhanced IL-1β turnover under conditions of Caspase-1 loss, ATP or nigericin-induced degradation of IL-18 was prevented by Caspase-1 and NLRP3 deficiency (Fig. 3b, c and Supplementary Fig. 3a). Moreover, consistent with ubiquitin-mediated proteasomal targeting, rather than inflammasome-mediated proteolysis, treatment of LPS-primed BMDMs with ATP and nigericin augmented the levels of ubiquitylated IL-1β in macrophages lacking Caspase-1, NLRP3, or ASC (Fig. 3d and Supplementary Fig. 3b). Consequently, NLRP3 inflammasome activating agents appear to have the capacity to signal precursor IL-1β ubiquitylation and proteasomal targeting independent of Caspase-1, which may act as an additional safeguard to limit inflammatory responses upon cellular stress.

**IL-1β K133 ubiquitylation promotes IL-1β degradation in 293 T cells.** Our results demonstrate that IL-1β undergoes ubiquitylation in response to TLR priming and NLRP3 activating stimuli. However, the site(s) for this key post-translational modification remains unclear. To evaluate which residues of IL-1β are targeted

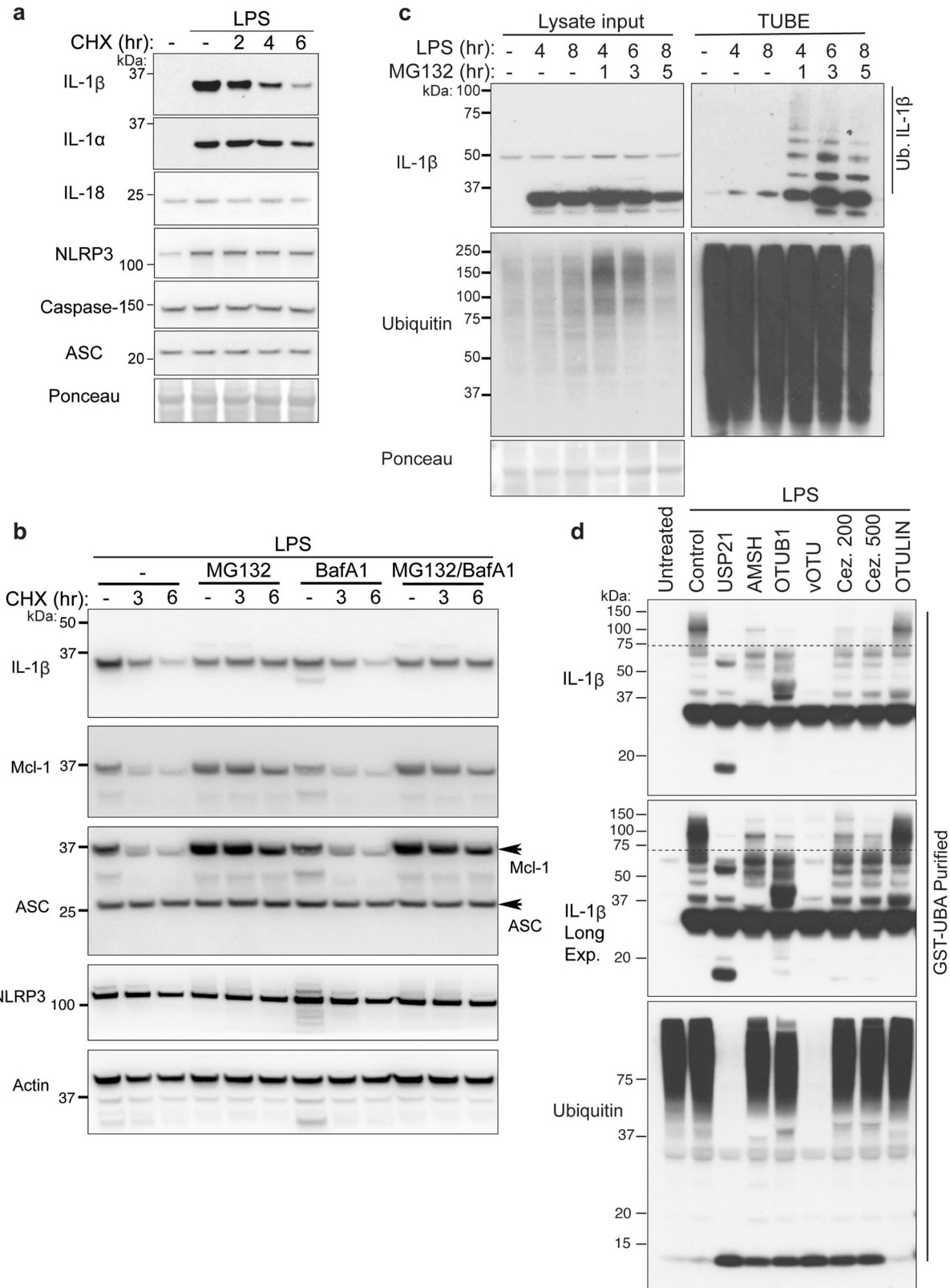

for ubiquitylation, we expressed N-terminal FLAG-tagged full-length IL-1β in 293T cells, which retains the ability to undergo ubiquitylation and proteasomal targeting (Fig. S4). We enriched the pool of modified FLAG-tagged IL-1β by treating cells with MG132 to limit proteasomal turnover, and subjected FLAG purified IL-1β to trypsin digestion and mass spectrometry. This analysis identified that lysine 133 (K133) of IL-1β was targeted for

ubiquitylation (marked by a Lys-ε-Gly-Gly (diGly) linkage), while the adjacent serine (S134) was phosphorylated (Fig. 4a, b). Notably, these post-translational modifications are located within the C-terminal bioactive region of the IL-1β fragment that is generated following Caspase-1 cleavage at D117 (Fig. 4c), and, importantly, are conserved between mouse and human IL-1β (Fig. 4d).

**Fig. 2 IL-1β is modified by K48-linked, K63-linked, and K11-linked ubiquitin chains and is targeted for proteasomal destruction. a** BMDMs were stimulated with Lipopolysaccharide (LPS, 100 ng/ml) for 3 h and Q-VD-OPh (20 μM) added in the last 30 min of priming. Cells were then treated with the protein synthesis inhibitor cycloheximide (CHX, 20 μg/ml) for up to 6 h, as indicated, and total cell lysates analyzed for the indicated proteins by immunoblot. One of two experiments. **b** BMDMs were stimulated with LPS (100 ng/ml) for 2.5 h; with Q-VD-OPh (20 μM) added in the last 30 min alongside, as indicated, the proteasome inhibitor MG132 (20 μM) and the inhibitor of lysosome function bafilomycin A1 (BafA1 300 nM). Cells were then treated with cycloheximide (CHX, 20 μg/ml) for up to 6 h as specified. Total cell lysates were subjected to immunoblot for the indicated proteins. One of three experiments. **c** BMDMs were treated with LPS (50 ng/ml) for up to 8 h, and MG132 (20 μM) added for the specified times. Ubiquitylated proteins were isolated from cell lysates using Tandem Ubiquitin Binding Entity (TUBE) purification, and cell lysate input and TUBE purified ubiquitylated (Ub.) proteins analyzed by immunoblot. One of three experiments. **d** iBMDMs were treated with LPS for 6 h and ubiquitylated proteins isolated by Glutathione S-transferase-Ubiquitin Associated domain (GST-UBA) purification. Specific ubiquitin chain linkages were cleaved via treatment with the indicated DUBs: USP21 (total ubiquitin), AMSH (K63-linked), OTUB1 (K48-linked), vOTU (total ubiquitin except M1), 200 and 500 nM Cezanne (Cez., K11-linked) and OTULIN (M1 linear), and immunoblots performed for the indicated proteins. One of three experiments.

Interestingly, structural analysis of IL-1β[26] revealed that K133 also forms a salt bridge with aspartate 129 (D129) (Fig. 4e). For this reason, we not only generated IL-1β point mutations to examine the role of IL-1β K133 ubiquitylation on IL-1β turnover, but also the impact of the K133:D129 electrostatic interaction on protein function. Importantly, both N-terminal FLAG-tagged and untagged WT IL-1β expressed in 293T cells were turned over at comparable rates to endogenous IL-1β in macrophages, and were degraded between 2 and 7 h following exposure to the protein synthesis inhibitor CHX (Figs. 2a, b, 5a and Supplementary Fig. 5a). This turnover of WT IL-1β following CHX treatment was rescued by proteasome inhibition using MG132 (Fig. 5b and Supplementary Fig. 5b). In line with endogenous IL-1β degradation being unrelated to Caspase activity (Fig. 3), mutation of the Caspase-1 and Caspase-8 cleavage site in IL-1β (IL-1β D117A; Fig. 4c) had no impact on IL-1β turnover (Supplementary Fig. 5b). Analysis of an IL-1β K133R mutant (to block K133 ubiquitylation but not the K133:D129 electrostatic interaction) revealed that the loss of IL-1β K133 ubiquitylation limits, but does not prevent, IL-1β turnover following CHX treatment (Fig. 5a and Supplementary Fig. 5a, c). Remarkably, abrogation of the K133:D129 electrostatic interaction (IL-1β K133A and IL-1β D129A point mutations) increased IL-1β stability even further (Fig. 5a and Supplementary Fig. 5c), raising the interesting possibility that the integrity of the K133:D129 electrostatic interaction is also important for proteasomal targeting. Of note, the *Il1b* containing plasmid we used in these experiments (pMIGRMCS) also harbors GFP downstream of an internal ribosome entry site (IRES), thereby allowing assessment of plasmid levels amongst cells transfected with different *Il1b* cDNAs. Analysis of GFP expression demonstrated that differences in the IL-1β mutant protein levels did not result from variances in the amount of plasmid transfected, as GFP expression was comparable between all *Il1b* constructs (Supplementary Fig. 5b, c).

**Loss of IL-1β K133 ubiquitylation does not impede IL-1β activation or release.** To explore if IL-1β K133 ubiquitylation and/ or the K133:D129 electrostatic interaction also impacted Caspase-1-mediated cleavage and release of IL-1β, we co-transfected 293T cells, which lack GSDMD expression[27], with *Caspase-1* and *Il1b* cDNAs. Caspase-1 expression resulted in WT IL-1β cleavage into its active p17 fragment, which was detected in both cell lysates and supernatants and, as expected, its processing was blocked by treatment with the pan-caspase inhibitor ZVAD-fmk (Fig. 5c). Loss of IL-1β K133 ubiquitylation (IL-1β K133R) did not prevent Caspase-1-mediated cleavage nor the release of IL-1β into the cellular supernatant (Fig. 5c). In contrast, abrogation of the IL-1β K133:D129 salt bridge (IL-1β K133A and D129A mutations) curtailed Caspase-1-mediated processing of IL-1β, resulting in reduced secretion of the bioactive fragment (Fig. 5c, d). Examination of GFP

levels demonstrated equivalent IRES-driven GFP expression from all *Il1b* containing plasmids (Fig. 5c). Similar to Caspase-1, expression of Caspase-8, which we and others have documented to process IL-1β[28–32], also resulted in WT IL-1β and IL-1β K133R secretion, and this was reduced upon expression of the IL-1β K133A mutant that abolishes the K133:D129 electrostatic interaction (Fig. 5e). In contrast, mutation of the IL-1β phosphorylation site we identified, S134 (Fig. 4), to either mimic non-phosphorylated (S134A) or phosphorylated (S134E) IL-1β, had no impact on either IL-1β turnover (Supplementary Fig. 5d), or on Caspase-1 or Caspase-8-mediated release (Fig. 5d, e). Notably, Caspase-1 and Caspase-8 expression did not result in any increase in cell death (Supplementary Fig. 5e, f), in agreement with studies suggesting that in some cell types, such as 293Ts, cell death-independent secretory pathways for activated IL-1β exist[27,33,34].

**Loss of IL-1β K133 ubiquitylation in mice increases precursor IL-1β.** To determine if the increased IL-1β levels associated with mutation of the K133 ubiquitylation site (Fig. 5) is physiologically relevant, we generated point mutant IL-1β K133R mice via CRISPR/Cas9 gene targeting (*Il1b*^K133R/K133R^; Supplementary Fig. 6a, see "Methods" section). Intriguingly, the resultant *Il1b*^K133R/K133R^ mice displayed no apparent phenotype compared to WT control animals. Moreover, LPS treatment of BMDMs from WT and *Il1b*^K133R/K133R^ mice also resulted in equivalent induction of *Tnf* and *Il1b* mRNA (Supplementary Fig. 6b, c). In comparison, examination of protein stability in response to LPS or Pam3Cys priming and CHX treatment to block protein synthesis, revealed that *Il1b*^K133R/K133R^ BMDMs exhibited elevated precursor IL-1β levels and reduced turnover compared to WT BMDMs, despite displaying similar levels of NLRP3 and Caspase-1, and comparable rates of Mcl-1 turnover (Fig. 6a and Supplementary Fig. 6d). These results confirm that ubiquitylation of IL-1β at K133 is required for efficient proteasomal degradation of TLR-induced IL-1β.

Next, to examine if Caspase-1-induced activation and release of IL-1β was altered in *Il1b*^K133R/K133R^ cells, we primed BMDMs with LPS and stimulated cells with the NLRP3 inflammasome activators ATP, nigericin, alum, or the inhibitor of apoptosis (IAP) protein antagonist Compound A (Cp. A). Compound A induces IL-1β activation in myeloid cells via direct Caspase-8-mediated proteolysis and triggering of NLRP3 inflammasome activity[29,35]. Consistent with increased IL-1β levels upon loss of IL-1β K133 ubiquitylation, ATP, and nigericin treatment of LPS primed *Il1b*^K133R/K133R^ macrophages resulted in increased IL-1β, but not TNF, detection in the cell supernatants when measured by ELISA (Fig. 6b, c). Similarly, alum or Cp.A treatment triggered enhanced IL-1β release in *Il1b*^K133R/K133R^ BMDMs, compared to control cells (Supplementary Fig. 6e). As previously reported[36], we observed that macrophages stimulated with NLRP3 activating agents in Opti-MEM media increased overall IL-1β release

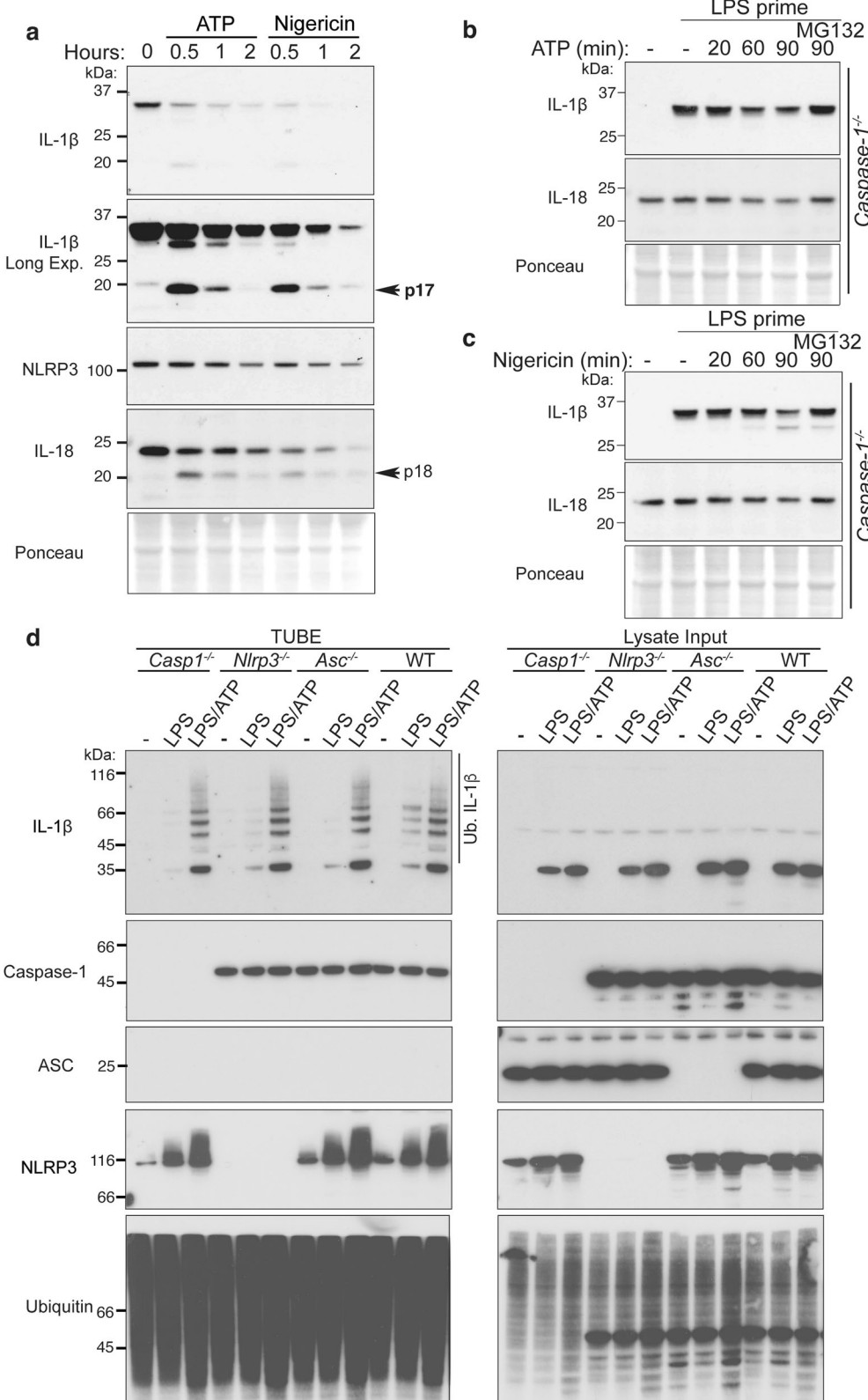

**Fig. 3 ATP and nigericin can augment proteasomal-mediated IL-1β degradation. a** BMDMs were primed with Lipopolysaccharide (LPS, 50 ng/ml) for 3 h and stimulated with ATP (5 mM) and Nigericin (10 μM) for up to 2 h. Total cell lysates were analyzed by immunoblot. One of three experiments. **b**, **c** Caspase-1⁻/⁻ BMDMs were primed with LPS (50 ng/ml) for 3 h, and in the last 30 min of priming treated with MG132 (20 μM), as indicated. Cells were treated with **b** ATP (5 mM) and **c** Nigericin (10 μM) for up to 90 min. Immunoblots were performed on total cell lysates to detect the specified proteins. One of three experiments. **d** WT, Nlrp3⁻/⁻, Asc⁻/⁻ and Caspase-1⁻/⁻ BMDMs were primed with LPS (50 ng/ml) for 3 h and stimulated with ATP for 20 min. Ubiquitylated (Ub.) proteins were purified from cell lysates by Tandem Ubiquitin Binding Entities (TUBEs). Immunoblots were performed on cell lysates and TUBE isolated ubiquitylated proteins for the indicated proteins. One of two experiments.

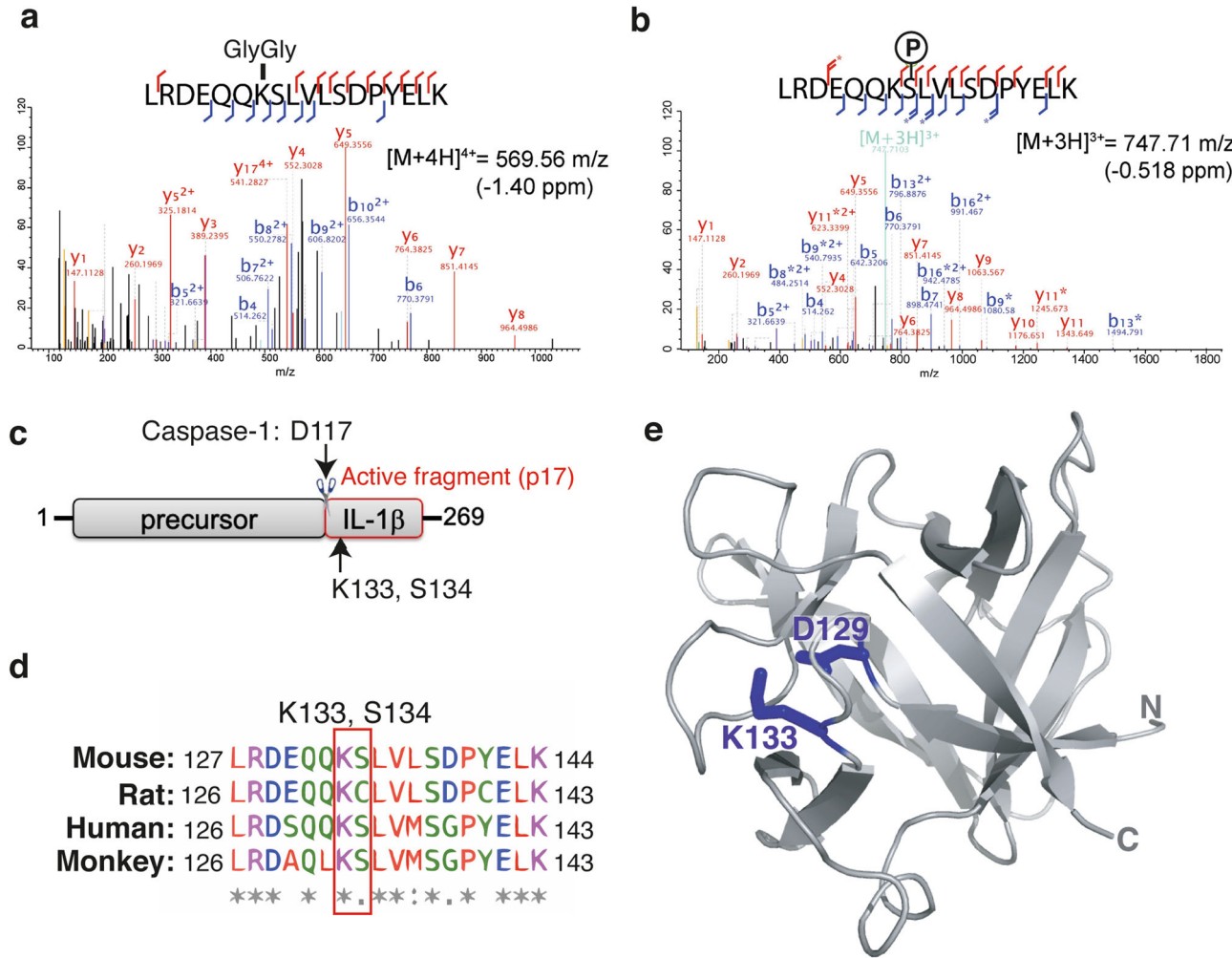

**Fig. 4 IL-1β lysine 133 is ubiquitylated and forms an electrostatic interaction with aspartate 129. a, b** N-terminal FLAG tagged IL-1β was expressed in 293T cells and treated with MG132 (20 μM) for 6 h. FLAG-immunoprecipitated IL-1β was run on a SDS-PAGE gel and in-gel trypsin digestion performed on SYPRO Ruby stained protein bands corresponding to the molecular weight of precursor IL-1β and modified IL-1β. Samples were then analyzed by high-resolution mass spectrometry to identify IL-1β modification sites, particularly ubiquitylation sites (marked by a K-ε-GG linkage). MS/MS spectra indicate that IL-1β was ubiquitylated on K133 and phosphorylated on S134. Stars indicate neutral loss fragmentation ions, clearly demonstrating phosphate localization at serine 134. **c** Schematic of precursor IL-1β detailing the caspase-1 and -8 cleavage site (D117) and the location of K133 and S134 post-translational modifications within the C-terminal bioactive region. **d** Sequence alignment showing conservation of the region of IL-1β harboring K133 and S134 amongst mouse, rat, human and monkey species. **e** Structure of IL-1β (PDB: 2MIB) documenting the K133 and D129 electrostatic interaction.

compared to cells maintained in L929 media and, interestingly, Opti-MEM media markedly augmented *Il1b*^K133R/K133R BMDM responses when compared to WT cells (Supplementary Fig. 6e). Consistent with ELISA data, immunoblot analysis revealed appreciably higher levels of cleaved, bioactive, 17 kDa IL-1β in cell supernatants of ATP or nigericin treated *Il1b*^K133R/K133R BMDMs when compared to WT cells (Fig. 6d). Importantly, levels of NLRP3, and Caspase-1 and IL-18 processing and activation, were comparable between WT and *Il1b*^K133R/K133R macrophages (Fig. 6d), confirming that differences were attributable to increased IL-1β levels. Mirroring these findings, *Il1b*^K133R/K133R neutrophils, purified from the bone marrow, displayed increased levels of precursor IL-1β upon IFNγ and LPS priming, and enhanced bioactive IL-1β release upon subsequent nigericin-induced NLRP3 inflammasome activation (Fig. 6e and Supplementary Fig. 6f).

**IL-1β K133 is not the only site targeted for ubiquitylation.** We next asked whether the ubiquitylation of IL-1β was impacted by loss of the IL-1β K133 site. Consistent with the DUB analysis (Fig. 2d),

immuno-precipitation of endogenous IL-1β from BMDMs treated with LPS and the proteasome inhibitor PS341 documented the presence of K11-linked, K48-linked, and K63-linked ubiquitin chains (Fig. 7a). Notably, all these ubiquitin linkage types were also detected on IL-1β derived from *Il1b*^K133R/K133R BMDMs. However, when taking into account the moderately elevated precursor IL-1β levels in *Il1b*^K133R/K133R macrophages, the observed ubiquitylation pattern indicates a possible reduction in IL-1β^K133R ubiquitylation, particularly its decoration with K11-linked ubiquitin chains (Fig. 7a). Regardless, and consistent with this analysis, TUBE purification isolated similar amounts of ubiquitylated IL-1β species derived from WT and *Il1b*^K133R/K133R BMDMs (Supplementary Fig. 7a). Therefore, IL-1β must be ubiquitylated on other residues, in addition to K133.

**Ubiquitylated IL-1β is not efficiently cleaved by caspase-1.** We next asked if cleaved bioactive IL-1β was ubiquitylated. BMDMs were LPS primed then stimulated with the NLRP3 trigger nigericin (to activate IL-1β and concentrate the p17 fragment in the cell supernatant). TUBE purification was subsequently

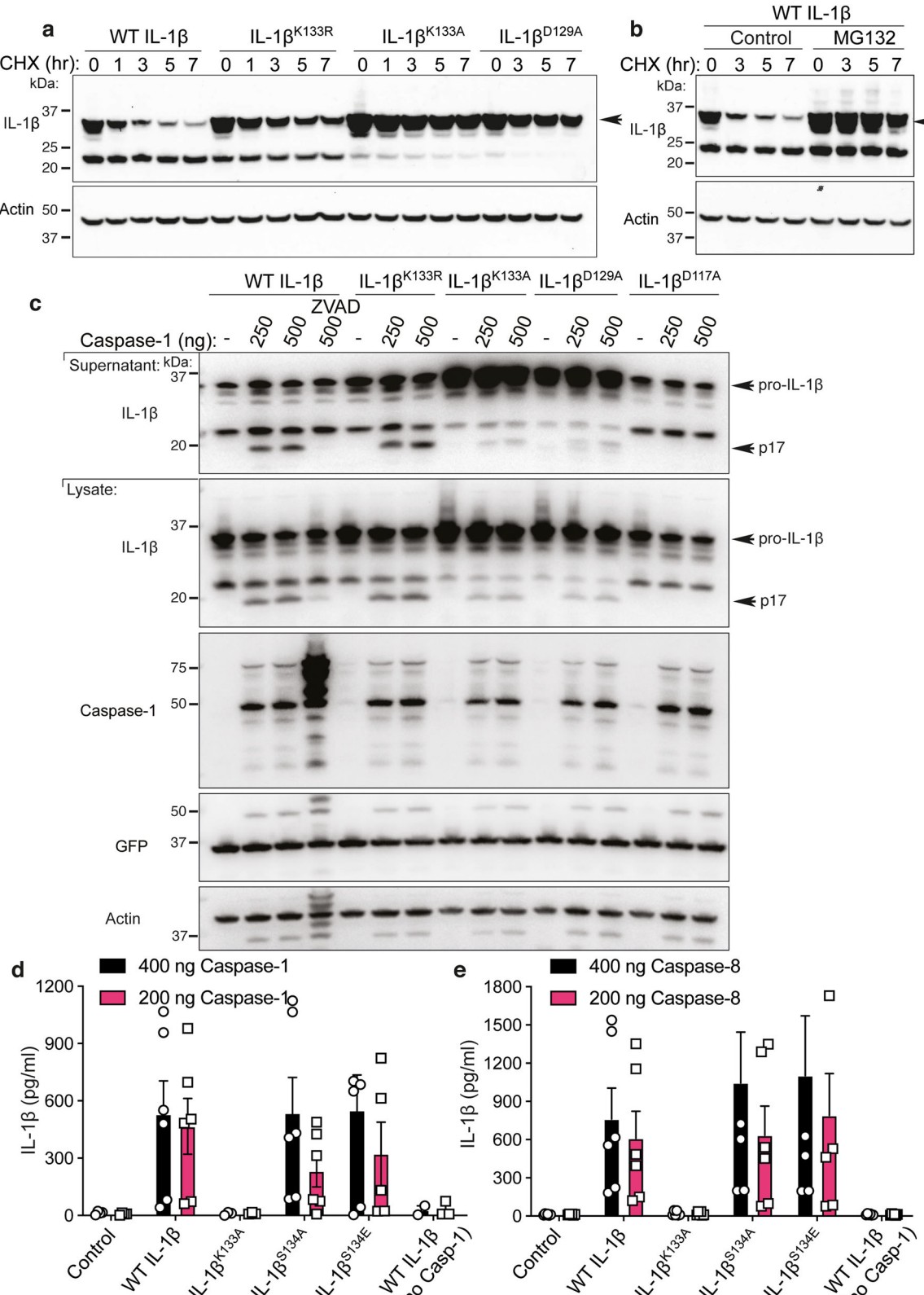

performed to isolate the ubiquitylated proteome from equivalent amounts of cell lysates and cell supernatants. Inflammasome-activated IL-1β was, at best, very weakly detected in TUBE purified cell supernatant (Supplementary Fig. 7a, LPS/nigericin supernatant TUBE). Consistent with this, when compared to the levels of activated IL-1β observed in cell supernatants, active IL-1β was not depleted in the post-TUBE cell supernatant

(i.e., samples from which ubiquitylated proteins had been removed) (Supplementary Fig. 7a). Therefore, bioactive IL-1β released from cells is unlikely to be decorated by ubiquitin chains.

Our data indicated that activated and released IL-1β is not modified by ubiquitin, despite this bioactive IL-1β fragment containing several lysine residues, including K133. This raised the interesting possibility that ubiquitylated IL-1β may not be

**Fig. 5 Loss IL-1β K133 ubiquitylation or the IL-1β K133:D129 salt bridge stabilizes IL-1β. a** WT *Il1b*, *Il1b*[K133A] and *Il1b*[D129A] cDNAs were expressed in 293T cells and 48 h after transfection cells exposed to cycloheximide (CHX, 20 μg/ml) for up to 7 h. Cell lysates were immunoblotted for IL-1β levels. The arrow indicates pro-IL-1β. Actin is included as a loading control. One of three experiments. **b** WT *Il1b* cDNA was transfected into 293T cells and after 48 h cells were cultured with MG132 (20 μM) as indicated for 15 min prior to cycloheximide (CHX, 20 μg/ml) treatment for up to 7 h. Immunoblots were performed on total cell lysates for IL-1β protein levels. The arrow indicates pro-IL-1β. Actin is included as a loading control. One of two experiments. **c** 293T cells expressing WT IL-1β, IL-1β[K133R], IL-1β[K133A], IL-1β[D129A] and IL-1β[D117A] were co-transfected with Caspase-1 (250 or 500 ng) in the presence or absence of the pan-caspase inhibitor Z-VAD-fmk (50 μM) for 24 h. Immunoblots were performed on cell supernatants and total cell lysates for the indicated proteins. One of 2–3 experiments. **d, e** 293T cells expressing WT IL-1β, IL-1β[K133R], IL-1β[K133A], IL-1β[S134A] (phospho-ablating) and IL-1β[S134E] (phospho-mimetic) were co-transfected with 200 and 400 ng of **d** Caspase-1 or **e** Caspase-8, as indicated, for 24 h. IL-1β secretion was analyzed by ELISA. Data are the mean + S.D of three independent experiments.

efficiently processed by Caspase-1. Therefore, we analyzed whether TUBE purification of ubiquitylated IL-1β was altered by recombinant Caspase-1 treatment, which would indicate that it was able to be processed by Caspase-1. Conversely, if ubiquitylated IL-1β levels remained similar this would indicate that Caspase-1 cannot recognize ubiquitylated IL-1β. BMDMs were stimulated with LPS and the proteasome inhibitor PS341 to allow the accumulation of endogenous ubiquitylated precursor IL-1β. Subsequently, cell lysates were generated and treated with recombinant Caspase-1 followed by TUBE purification. Remarkably, ubiquitylated precursor IL-1β purification was identical between untreated and Caspase-1 treated samples (Fig. 7b). Furthermore, and consistent with analysis of released IL-1β (Supplementary Fig. 7a), bioactive p17 IL-1β ubiquitylation was not detected, despite extensive Caspase-1 processing of input cell lysate precursor IL-1β (Fig. 7b). The fraction of non-modified IL-1β consistently observed to co-purify with TUBE-isolated ubiquitylated IL-1β was also not targeted for Caspase-1-mediated proteolysis (Fig. 7b). This suggests that ubiquitylated IL-1β complexes cannot undergo Caspase-1 processing. The substantial processing and thus depletion of cell lysate (input) precursor IL-1β by recombinant Caspase-1 further indicates that the majority of IL-1β in a cell at any one time is not ubiquitylated (Fig. 7b). We therefore suggest that the K11-linked, K48-linked, and K63-linked ubiquitin chains that decorate precursor IL-1β are likely involved in both its proteasomal targeting and also act to prevent Caspase-1-mediated processing.

**Loss of IL-1β K133 ubiquitylation increases bioactive IL-1β levels in vivo**. To address whether *Il1b*[K133R/K133R] mice phenocopy in vitro responses and display elevated IL-1β due to an increased stabilization of IL-1β, we intra-peritoneally injected WT control and *Il1b*[K133R/K133R] mice with LPS (100 μg). Notably, levels of IL-1β were significantly enhanced in both the serum and peritoneal lavage fluid of *Il1b*[K133R/K133R] mice compared to WT control animals (Fig. 7c). In contrast, serum and peritoneal lavage fluid levels of TNF and IL-6 were comparable between groups (Fig. 7d, e). Western blot analysis of peritoneal lavage fluid also documented a marked increase in both precursor and bioactive IL-1β in *Il1b*[K133R/K133R] mice, relative to WT animals, while NLRP3 levels remained similar (Fig. 7e and Supplementary Fig. 7b). Therefore, ubiquitylation of IL-1β at K133 also acts to limit LPS-induced inflammation in vivo by promoting IL-1β turnover, thereby reducing its ability to be activated by inflammasomes.

## Discussion

Our findings provide evidence that cellular IL-1β is rapidly turned over by ubiquitylation and proteasomal targeting and that, upon its decoration with ubiquitin chains, precursor IL-1β becomes inaccessible to caspase-1 cleavage. We demonstrate that the proteasomal targeting of IL-1β plays an important role in regulating precursor IL-1β levels upon inflammasome priming and is also triggered by inflammasome activating agents independent of Caspase-1 activity. Moreover, our biochemical and genetic analysis, including the generation of *Il1b*[K133R/K133R] mice, highlight that IL-1β K133 ubiquitylation limits IL-1β levels and therefore Caspase-1-mediated activation both in vitro and in vivo. Thus, in addition to transcriptional and Caspase-mediated regulation of IL-1β, ubiquitin-mediated post-translational control of IL-1β critically determines its inflammatory capacity.

The post-translational regulation of the inflammasome machinery has received substantial attention in recent years, including ubiquitylation, SUMOylation, and phosphorylation of inflammasome sensor proteins, such as NLRP3, as well as the inflammasome adapter ASC and caspase-1[37–41]. Our TUBE purification and analysis of ubiquitylated inflammasome components suggest that only NLRP3 and IL-1β are robustly decorated by ubiquitin chains upon TLR-mediated inflammasome priming. NLRP3 activating agents have been proposed to trigger the deubiquitylation of NLRP3, via the DUBs murine BRCC3 and human BRCC36 that form the BRCC36 isopeptidase complex (BRISC), to promote inflammasome formation[38,42,43]. Emerging studies suggest ubiquitylation can regulate IL-1β, although reports on its biological function, and relevance to not only primary immune cells but also in vivo IL-1β responses, are divergent and remain unclear. For example, based on experiments in 293T cells, IL-1β ubiquitylation has been suggested to allow for Caspase-1-mediated cleavage-induced activation[17], while another study reported that the ubiquitylation of pro-IL-1β in colonic epithelial cells is required for cell death-independent secretion[44].

The degradation of inflammasomes and/or IL-1β has been reported to occur by either lysosomal or proteasomal targeting[45–48]. While we verify that inflammasome priming (or precursor IL-1β expression) triggers the rapid proteasomal turnover of IL-1β, blockade of lysosomal function had no impact on its levels. Moreover, in our hands, both the ubiquitylation and proteasomal targeting of IL-1β was enhanced by NLRP3 activating agents, ATP and nigericin, independent of Caspase-1. This suggests that these cellular stressors can activate parallel pathways leading to inflammasome activation and IL-1β degradation. In this regard, it is interesting that NLRP3-independent inflammasome stimuli, such as the AIM2 ligand poly(dA:dT) and the NLRC4 activator flagellin, have also been reported to increase levels of ubiquitylated IL-1β[49]. It is also notable that IL-1β exhibits a rapid turnover compared to other inflammasome components, such as NLRP3, Caspase-1, and ASC, that are relatively long-lived proteins. This highlights the cellular requirement to reduce the signaling capacity of IL-1β independent of the upstream inflammasome machinery. This uncoupling may reflect the fact that several proteolytic enzymes, such as Caspase-8, granzymes and neutrophil elastases, can also process IL-1β into a bioactive fragment[28,31,32,50–52]. Consequently, tight transcriptional and post-translational regulatory control of IL-1β is important for limiting damaging inflammation.

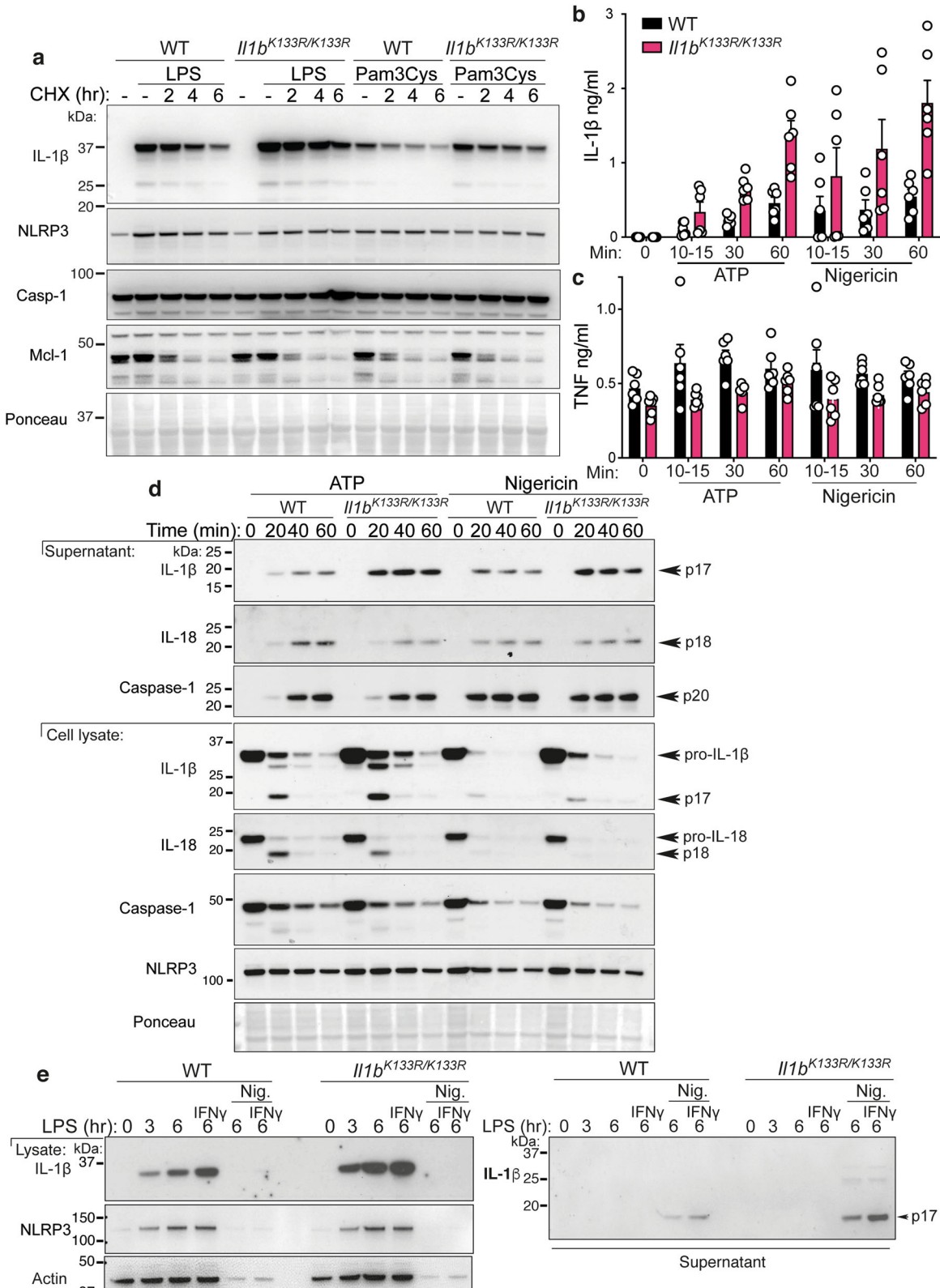

Our mass-spectrometry analysis identified IL-1β K133 as being targeted for ubiquitylation. Consistent with IL-1β K133 ubiquitylation contributing to its proteasomal targeting, IL-1β K133R expressing 293T cells, as well as macrophages and neutrophils derived from *Il1b*<sup>K133R/K133R</sup> mice, expressed increased levels and reduced turnover of precursor IL-1β. As such, more bioactive IL-1β was generated and released from *Il1b*<sup>K133R/K133R</sup> macrophages,

neutrophils and mice upon NLRP3 inflammasome activation, both in vitro and in vivo. Interestingly, analysis of the structure of IL-1β revealed a previously unappreciated electrostatic interaction between IL-1β K133 and D129. Abrogation of this salt bridge via generation of IL-1β K133A, or the opposing IL-1β D129A mutant, resulted in the same biological outcomes; stabilization and reduced turnover of IL-1β. The IL-1β K133:D129 electrostatic

**Fig. 6 Loss of K133 ubiquitylation in mice stabilizes IL-1β and increases IL-1β activation. a.** WT and $Il1b^{K133R/K133R}$ BMDMs were primed with Lipopolysaccharide (LPS, 50 ng/ml) or Pam3Cys-Ser-(Lys)4 (Pam3Cys, 500 ng/ml) for 3 h with Q-VD-OPh (20 μM) added in the last 30 min of priming. Cells were then treated with the protein synthesis inhibitor cycloheximide (CHX, 20 μg/ml) for up to 6 h. Total cell lysates were analyzed for the indicated proteins by immunoblot. One of four experiments. **b, c** WT and $Il1b^{K133R/K133R}$ BMDMs were primed with LPS (50 ng/ml) for 3 h and then treated with ATP (5 mM) and Nigericin (10 μM) for the times indicated in Opti-MEM media. **b** IL-1β and **c** TNF levels were measured in cell supernatants by ELISA. Data are the mean + SD of two independent experiments combined using three mice of each genotype. **d** WT and $Il1b^{K133R/K133R}$ BMDMs were primed with LPS (50 ng/ml) for 3 h and treated with ATP (5 mM) and Nigericin (10 μM) for up to 60 min in Opti-MEM media. Cell supernatants and lysates were analyzed by immunoblot for the indicated proteins. One of three experiments. **e** WT and $Il1b^{K133R/K133R}$ neutrophils sorted from bone marrow were primed with IFNγ (50 ng/ml, 1 h) and then treated with LPS (100 ng/ml) for a further 3 h. Cells were then stimulated with Nigericin (10 μM) for up to 6 h, as indicated. Cell supernatants and lysates were analyzed by immunoblot for the specified proteins. One of three experiments.

interaction was also essential for efficient IL-1β cleavage by Caspase-1. Therefore, our findings showing that ubiquitylated IL-1β, akin to the IL-1β K133A or D129A mutants, is inefficiently cleaved by Caspase-1 raises the interesting possibility that IL-1β K133 ubiquitylation impacts the integrity of the K133:D129 electrostatic interaction to limit caspase-1 processing. Collectively, when combined with our inability to detect ubiquitylated bioactive IL-1β, these findings demonstrate that activated IL-1β is likely derived from non-ubiquitylated pools of precursor IL-1β.

The K11-ubiquitin, K63-ubiquitin, and K48-ubiquitin linkage types we identified on modified precursor IL-1β are consistent with its proteasomal targeting. Although K63-linked ubiquitin chains have mainly been associated with altering protein complex formation, trafficking, or signaling, evidence also exists that they can facilitate proteasomal targeting[22–24,53,54]. The presence of K63-linked ubiquitin chains on purified IL-1β has been reported by others, and it has been suggested that the DUB A20 represses IL-1β ubiquitylation[17]. However, because A20 deficiency increases TLR-induced precursor IL-1β levels in a dominantly MyD88-dependent manner (i.e., transcriptionally), enhanced IL-1β ubiquitylation upon A20 loss may reflect increased IL-1β substrate availability[17,55]. In keeping with this idea, the IL-1β ubiquitylation we observed in $Il1b^{K133R/K133R}$ macrophages may also be impacted by its stabilization and overall increased accessibility for ubiquitin modification on alternate residues.

More recently, deletion of the DUB POH1 in macrophages was shown to increase IL-1β activation following treatment with NLRP3 stimuli[49]. Mechanistically, it was suggested that POH1 acts to remove K63-linked ubiquitin chains from precursor IL-1β K133 (and/or K247) to reduce its ability to be processed upon inflammasome triggering. Supporting this notion, expression of IL-1β K133R in 293T cells limited IL-1β modification by overexpressed K63-linked ubiquitin chains, and also reduced its ability to be cleaved following the transfection of caspase-1[49]. These data do not agree with our findings showing that Caspase-1 cleavage of IL-1β K133R in 293T cells occurs normally, and that it is only impeded when the K133:D129 salt bridge is disrupted (i.e., D129A or K133A mutations). Our findings are further supported by our in vitro and in vivo data demonstrating enhanced, not reduced, IL-1β levels and inflammasome processing in $Il1b^{K133R/K133R}$ mutant mice and macrophages. Intriguingly, POH1 is a subunit of the 19S proteasome and removes ubiquitin from proteins to allow these substrates to enter the catalytic barrel of the 20S proteasome to be degraded[56–59]. It is tempting to speculate that POH1 loss may enhance IL-1β activation due to inefficient proteasomal clearance.

Contrary to other cytokines targeted into the conventional secretory pathway, leaderless IL-1β is exposed to cytosolic post-translational modifications. Our study demonstrates that ubiquitin-mediated degradation of cytosolic IL-1β by the proteasome acts as an important checkpoint to control levels of precursor IL-1β upon inflammasome priming, and also upon sensing of inflammasome activating stimuli. While IL-1β K133 modification by ubiquitin is involved in limiting IL-1β levels, and

hence activity, the modification of other residues also seem to contribute to its proteasomal degradation. Identifying the ubiquitin E3 ligases required for IL-1β turnover will further inform as to the roles of ubiquitin in regulating IL-1β levels, and its function in health and disease.

## Methods

**Mice.** All mice were housed at the Walter and Eliza Hall Institute of Medical Research (WEHI), Australia. Animal rooms were monitored to maintain suitable environmental conditions for the mice. Temperature: Animal rooms were maintained at approximately 21 °C with a band range +/−3 °C (from 18 to 24 °C). Light cycles: Timed 14/10 h light/dark cycle. Humidity: monitored within the facility and mirrored levels outside the building. Preferred ranges were between 40–70%. Ranges that fell beyond these measures triggered increased monitoring of cage conditions within the facility. Exhaust: Air handling units were monitored and alarmed by Facilities Management to provide 16 Air changes per hour. The WEHI Animal Ethics Committee approved all experiments in accordance with the NHMRC Australian code for the care and use of animals for scientific purposes. Wildtype (WT) C57BL/6 mice were obtained from WEHI animal supplies (Kew, Australia), and $Nlrp3^{-/-}$[60], $Asc^{-/-}$[61], and $Caspase-1^{-/-}$[62] mice were kindly provided by Seth Masters (WEHI). These mice were either generated on a C57BL/6 background ($Nlrp3^{-/-}$) or using strain 129 embryonic stem cells ($Caspase-1^{-/-}$, $Asc^{-/-}$) followed by backcrossing onto a C57BL/6J background for at least ten generations. $Il1b^{K133R/K133R}$ mice harboring a lysine to arginine mutation at amino acid position 133 of IL-1β were generated on a C57BL/6J background by the MAGEC facility (WEHI). To generate $Il1b^{K133R/K133R}$ mice, 20 ng/μl of Cas9 mRNA, 10 ng/μl of sgRNA (CATATGGGTCCGACAGCACG) and 40 ng/μl of oligo donor (CTGCTGGTGTTGTGACGTTCCCATTAGACAACTGCACTA-CAGGCTCCGAGATGAACAACAAAGAAGTCTCGTGCTGTCGGACCCA-TATGAGCTGAAAGCTCTCCACCTCAATGGACAGAATATCAACCA) were injected into the cytoplasm of fertilized one-cell stage embryos derived from WT C57BL/6 breeders. The oligonucleotide donor sequence used to generate the $Il1b^{K133R}$ mutation is explained in Supplementary Fig. 6a. Viable founder mice were identified by next-generation sequencing. Targeted animals were backcrossed onto C57BL/6J animals for two generations to eliminate potential sgRNA off-target hits, and then subjected to a further round of next-generation sequencing. Heterozygous $Il1b^{K133R/+}$ mice were intercrossed to generate WT littermates and homozygous $Il1b^{K133R/K133R}$ knockin mouse lines. Mice were routinely genotyped using genomic DNA extracted from tail biopsies using the Direct PCR Lysis tail reagent (Viagen) supplemented with 5 mg/ml proteinase K (Worthington), in accordance with the manufacturer's instructions. Three primers (also see Supplemental Table 1 for primers used in this study) were used to amplify, and distinguish between WT $Il1b$ and $Il1b^{K133R}$ allele's via two PCRs; 5′ TCCGAGATGAACAACAAAAAAGC 3′ ($Il1b$, forward), 5′ TCCGAGATGAA-CAACAAAGAAGT 3′ ($Il1b^{K133R}$, forward) and 5′ GTTATCCCTA-TATGACTGGGCTGG 3′ ($Il1b$ common, reverse).

**Endotoxin model.** Female $Il1b^{K133R/K133R}$ and littermate WT control mice aged 6–8 weeks were intra-peritoneally injected with 100 μg LPS (Ultra-pure, Invivogen) and euthanized 2 h later. Cardiac bleeds and peritoneal lavages were performed to collect serum and peritoneal fluid, respectively, and samples were stored at −80°C until analysis.

**Myeloid cell culture.** To generate macrophages, bone marrow cells harvested from femoral and tibial bones from 7–12 week old mice of either sex and were cultured in Dulbecco's modified essential medium (DMEM) containing 10% fetal bovine serum (FBS, Sigma), 50 U/ml penicillin and 50 μg/ml streptomycin (complete media) and supplemented with 15–20% L929 cell conditioned medium for 6 days (37 °C, 10% $CO_2$). Unless otherwise indicated, BMDMs or iBMDMs were routinely plated at $4 \times 10^5$ cells/well in 24-well tissue culture plates (BD Falcon) or $1 \times 10^5$ cells/well in 96-well flat bottom tissue culture plates. To isolate neutrophils, bone marrow cells were subjected to red blood cell lysis and stained with fluorochrome-conjugated anti-mouse antibodies to CD11b (Mac-1, BD Biosciences 553310; 1:400 dilution) and/or Ly6G

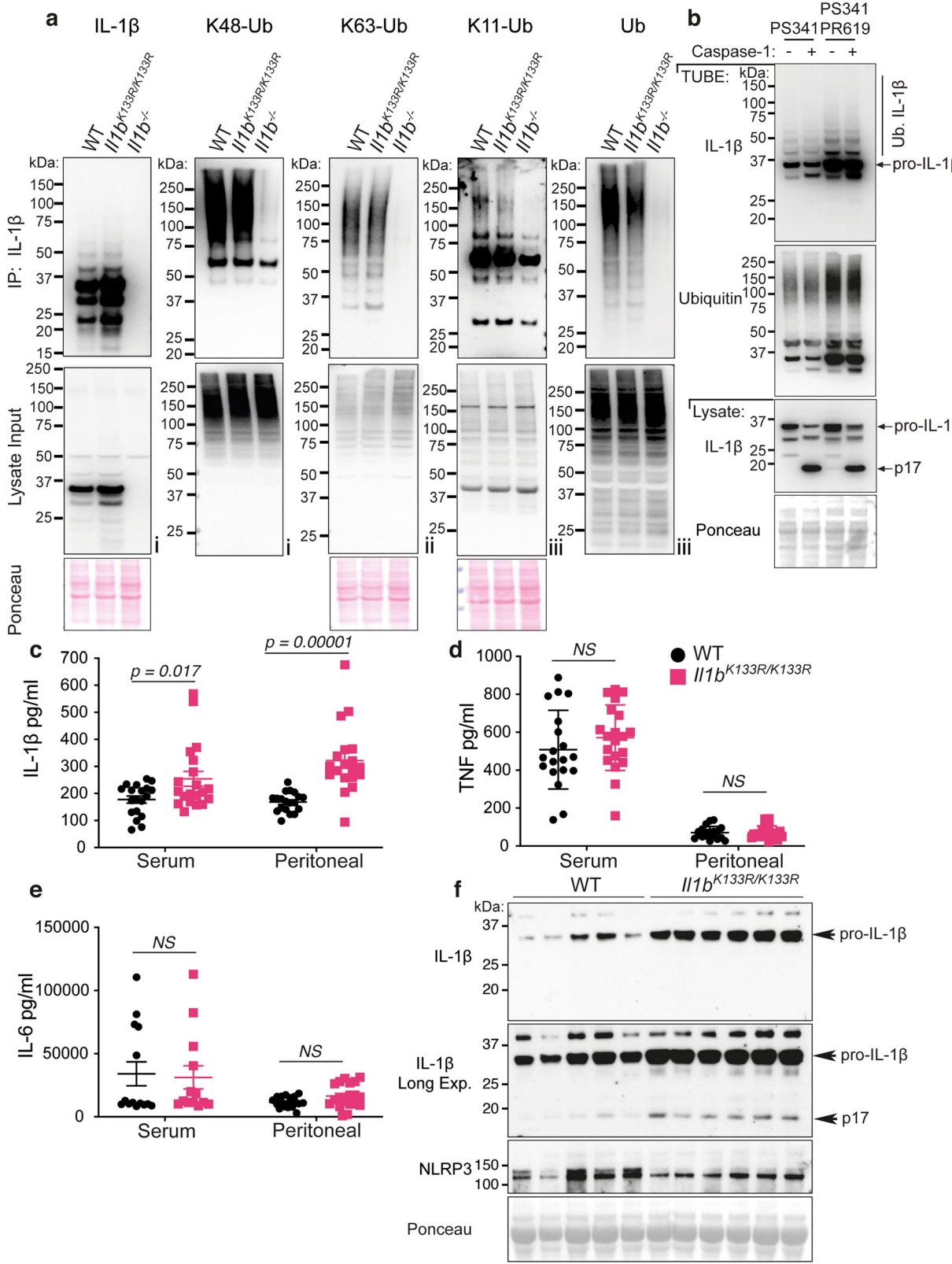

(1A8, Invitrogen, 12-9668-82; 1:400 dilution). Viable (PI⁻) neutrophils (CD11b⁺ and/or Ly6G⁺) were sorted using a BD FACSAria III sorter (WEHI). Purified neutrophils were plated at $2 \times 10^5$ per well in 96-well tissue culture plates. Macrophages and neutrophils were primed, where indicated with recombinant murine IFN-γ (50 ng/ml, R&D Systems) for 1 h (for neutrophil cultures) and LPS (50–100 ng/ml Ultrapure, InvivoGen) or Pam3Cys (500 ng/ml, InvivoGen) for 2–3 h, prior to stimulation, as indicated, with Cycloheximide (CHX, 20 μg/ml, Sigma-Aldrich), MG132

(20 μM, SIGMA-Aldrich), Bafilomycin A1 (BafA1, 150-300 nM, Sigma-Aldrich), QVD-OPh (20 μM, Biovision) ZVAD-fmk (20 μM, Z-VAD-FMK, R&D Systems), 911 (1 μM, Compound A, TetraLogic Pharmaceuticals), Nigericin (10 μM, Sigma-Aldrich), ATP (5 mM, Sigma-Aldrich) and Alum (300 μg/ml, ThermoScientific). At specified times supernatants were harvested for cytokine analysis. Supernatants and cells were prepared in reducing and denaturing sample buffer for immunoblot analysis.

**Fig. 7 LPS-induced IL-1β is increased in *Il1b*[K133R/K133R] mice in vivo. a** BMDMs derived from mice of the indicated genotypes were treated with LPS (100 ng/ml) for 3 h, followed by proteasome inhibition (PS341, 10 μM) for an additional 3 h. Biotin conjugated IL-1β antibody was used to purify IL-1β from cell lysates and its modification analyzed using the indicated ubiquitin (Ub) chain-specific antibodies. Cell lysates from the same experiment were run in triplicate with roman numerals on the bottom right of the blots indicating the membrane probed. One of two independent experiments. **b** BMDMs were primed with LPS (100 ng/ml) for 3 h followed by treatment for a further 3 h with PS341 (10 μM). Cells were subsequently lysed in buffer containing either PS341 (10 μM), or PS341 and PR619 (10 μM), as indicated. Recombinant caspase-1 (5 Units) was added and cell lysates incubated at 37 °C for 80 min. Subsequently, ubiquitylated proteins were purified by Tandem Ubiquitin Binding Entities (TUBEs) and analyzed by immunoblot. One of two independent experiments. **c–f** WT and *Il1b*[K133R/K133R] mice were injected intra-peritoneally with 100 μg of LPS. After 2 h serum and peritoneal fluid was harvested and analyzed by ELISA for levels of **c** IL-1β, **d** TNF, and **e** IL-6. Data are mean ± SEM, symbols represent individual mice, n ≥ 19 mice/genotype from three pooled independent experiments for **c**, **d** and n ≥ 14 for **e** from 2 to 3 pooled experiments. *P*-values were calculated using an unpaired two-tailed t-test for comparisons between genotypes. **f** Peritoneal fluid was analyzed by immunoblot for the indicated proteins. Each lane represents an individual mouse. n = 5–6 mice per genotype. One of two independent experiments (analysis of nine additional WT and 19 *Il1b*[K133R/K133R] mice are shown in Supplementary Fig. 7b).

**Complementation of *Il1b*[−/−] iBMDMs**. Cre-J2 immortalized *Il1b*[−/−] BMDMs were passaged in DMEM containing 10% fetal bovine serum (FBS, Sigma), 50 U/ml penicillin and 50 μg/ml streptomycin (complete media) and supplemented with 15–20% L929 cell conditioned medium. Cells were infected with retrovirus containing murine *Il1b* cloned into pMIGMCS (harbors GFP downstream of an IRES). Macrophages were subsequently sorted by flow cytometry for GFP expression and analyzed as indicated in the figure legends.

**Cell viability**. To evaluate cell viability, cells were harvested and propidium iodide (1–2 μg/ml, PI) uptake measured by flow cytometric analysis on a FACS Calibur instrument and Cell quest software (BD Biosciences). FACs data was analyzed using WEASEL version 2.7 software.

**Cytokine analysis**. IL-1β (R&D, DY401), TNF (Invitrogen, 5017331), and IL-6 (Invitrogen, LS88706477) ELISA kits were used according to the manufacturer's instructions.

**Immunoprecipitation**. For ubiquitin linkage-specific antibodies cells were washed with ice-cold phosphate buffered saline (PBS) and lysed with 500 μl of denaturing DISC lysis buffer (30 mM Tris-HCl (pH 7.4), 120 mM NaCl, 2 mM EDTA, 2 mM KCl, 1% Triton X-100, Roche complete protease inhibitor cocktail, 10 μM PR619, and 4 M urea). Lysates were pre-cleared with 100 μl of Protein G-sepharose slurry. Immunoprecipitation was carried out by adding 4 μg of specified polyubiquitin linkage-specific antibody (K48-linked polyubiquitin, Merck; 05-1307, clone Apu 2 and K63-linked polyubiquitin, Merck 05-1308, clone Apu 3), or anti-E25 (IgG isotype-matched control), to cleared cell lysates. Samples were rotated for 20 h at 4 °C, followed by a 1 h incubation with 40 μl of protein G-sepharose slurry to capture antigen-antibody complexes. The protein G beads were washed twice with DISC lysis buffer (without urea) and once with PBS. Each wash was carried out for 10 min while rotating at 4 °C. Protein complexes bound to the beads were eluted with 25 μl of 2× SDS sample buffer (50 mM Tris pH 6.8, 2% (v/v) SDS, 10% (v/v) glycerol, 0.01% (v/v) bromophenol blue, 0.05% (v/v) 2-mercaptoethanol) and analyzed by Western blot. For purification of endogenous IL-1β, 1 × 10[7] BMDMs were primed with LPS (100 ng/ml) for 3 h and then treated with PS341 (10 μM) for a further 3 h. Cells were lysed in 1000 μl DISC lysis buffer containing 0.1% SDS on ice for 20–30 min, diluted out to a concentration of 0.04% SDS, clarified by centrifugation, 4 μg of biotin conjugated anti-IL-1β (Biolgend; 503505) added to soluble lysate, and samples incubated overnight at 4 °C while rotating. IL-1β complexes were then isolated using Dynabeads M-280 Streptavidin (2 h, 4 °C), followed by four washes in DISC and elution of IL-1β in SDS sample buffer.

**Tandem ubiquitin binding entity and GST-UBA purifications**. Following specified stimulations, ice-cold PBS-washed BMDMs (10–12 × 10[6]) were harvested in 800–1000 μl of DISC lysis buffer (30 mM Tris-HCl (pH 7.4), 120 mM NaCl, 2 mM EDTA, 2 mM KCl, 1% Triton X-100, Roche complete protease-inhibitor cocktail, 2 mM NEM) and lysed on ice for 30 min. Lysates were cleared by centrifugation (14,000×g, 10 min) and endogenous ubiquitylated proteins were isolated from the soluble lysate at 4 °C for 3–20 h using 20 μl of packed agarose TUBE (TUBE1, Life sensors, performed according to the manufacturer's instructions) or 20 μl of GST-UBA beads (generated in-house). Following four washes in DISC lysis buffer, bound proteins were eluted using reducing and denaturing Western blot sample buffer for Western blot analysis.

**293 T cell IL-1β turn over and caspase-1 cleavage assays**. For IL-1β turnover assays, confluent 293T cells plated on 10 cm tissue culture plates were transfected using lipofectamine 2000 (Life Technologies) with 10 μg of murine IL-1β and point mutant variants (cloned into pMIGMCS) and after 24 h cells harvested and plated onto 24-well tissue culture plates and left to settle overnight. Cells were then treated with CHX (20 μg/ml) and MG132 (20 μM), as indicated, and total cell lysates

generated for immunoblot analysis. For the 293T cell caspase-1 cleavage assays, 10 cm tissue culture plates of confluent 293T cells were transfected with 10 μg murine *Il1b* and point mutant variants, as described above, and after 24 h cells harvested and plated onto 24-well tissue culture plates. These cells were then transfected with *Caspase-1* (in pcDNA3.1; 250 or 500 ng per well) and treated with Z-VAD-fmk (30 μM), as indicated. After 24 h cell supernatants and total cell lysates were harvested for ELISA and immunoblot analysis. Alternatively, for in vitro Caspase-1 cleavage of ubiquitylated IL-1β, 1 × 10[7] BMDMs were primed with LPS (100 ng/ml) for 3 h and then treated with PS341 (10 μM, MedChemExpress) for a further 3 h. Cells were lysed in caspase assay buffer containing 1 mM DTT, 1% Triton-X100, 10 μM PS341 and 10 μM PR619 for 15 min on ice followed by addition of 5 Units of active recombinant Caspase-1 (Biovision, 1181). This lysate was incubated at 37 °C for 80 min and samples subsequently diluted in ice-cold DISC buffer (containing 2 mM NEM and Roche complete protease-inhibitor cocktail), cleared by centrifugation, and ubiquitylated proteins isolated by TUBE purification.

**Mass spectrometry analysis**. 293T cells were plated in 15 cm petri dishes and grown to 90% confluency. FLAG-IL-1β was transfected using Lipofectamine 2000. After 48 h, cells were washed with ice-cold PBS and lysed with 700 μl of non-denaturing DISC lysis buffer (30 mM Tris-HCl (pH 7.4), 120 mM NaCl, 2 mM EDTA, 2 mM KCl, 1% Triton X-100, Roche complete protease inhibitor cocktail, 10 μM PR619 and 1 mM NEM) on ice for 30 min. Cell lysates were spun at ~13,000×g (4 °C) and the supernatant collected. A 50 μl slurry of M2 agarose anti-FLAG conjugated beads was added to the clarified cell lysate and rotated overnight at 4 °C. Subsequently, anti-FLAG beads were washed four times with DISC lysis buffer and bound FLAG-IL-1β eluted three times with 50 μl of elution buffer (0.5% SDS and 5 mM DTT) at 65 °C. Eluted proteins were run on NuPAGE Novex 4-12% Bis-Tris gel (Life Technologies). SYPRO Ruby staining was used to view the purified proteins, in-gel trypsin digestion performed, and digested peptides analyzed by high resolution mass spectrometry, as previously described[63], to identify modified IL-1β residues. Raw files consisting of high-resolution MS/MS spectra were processed with MaxQuant (version 1.5.2.8) for feature detection and protein identification using the Andromeda search engine[64]. Extracted peak lists were searched against the reviewed *Homo sapiens* (UniProt, March 2015; https://www.uniprot.org/proteomes/UP000005640) database containing murine IL-1β sequence (P10749), as well as a separate reverse decoy database (MaxQuant version 1.5.2.8) to empirically assess the false discovery rate (FDR) using strict trypsin specificity, allowing up to two missed cleavages. Modifications: Carbamidomethylation of Cys was set as a fixed modification, while oxidation of Met, phosphorylation (Ser, Thr, and Tyr) and ubiquitination (Lys) were set as variable modifications. The mass tolerance for precursor ions and fragment ions were 20 ppm and 0.5 Da, respectively. The mass spectrometry proteomics data have been deposited to the ProteomeXchange Consortium via the PRIDE partner repository with the dataset identifier PXD021305.

**In vitro ubiquitylation assays**. 293T cells were transfected (Lipofectamine 2000) with 10 μg of FLAG-IL-1β. After 24–48 h cells were washed with PBS and lysed in 800 μl of non-denaturing DISC buffer for 20 min. Cell lysate was clarified (14,000 rpm, 10 min) and then added to 20 μl of M2 agarose anti-FLAG beads (equilibrated in 1 ml of DISC lysis buffer) and incubated at 4 °C overnight while rotating. Anti-FLAG beads were washed 4× with 1 ml of DISC lysis buffer, then 2 × 1 ml with Ubiquitin assay buffer (40 mM Tris-HCL pH 7.5, 10 mM MgCl₂, 0.6 mM DTT) and eluted with FLAG peptide (50 μl, 200 μg/ml diluted in ubiquitin assay buffer) for 30 min while mixing at room temperature. Ubiquitylation assays were performed at 37 °C for 90 min using loaded E1 (~80 ng, 3 μl, Boston Biochem kit, K-995), E2 (1 μM, 1.2 μl, UbcH5a, Boston Biochem, E2-616), E3 (0.5 μM, 6 μl, His-cIAP1, Boston Biochem, E3-280) and 15 μl of purified FLAG-IL-1β in Ubiquitin assay buffer containing 2 mM ATP (total volume of 30 μl). The reactions were stopped by the addition of 5× reducing and denaturing western blot sample buffer and boiling for 5 min prior to immunoblot analysis.

**UbiCRest analysis**. Ubiquitylated IL-1β was isolated from iBMDMs that had been treated with LPS (50 ng/ml) for 6 h via GST-UBA purification. Subsequently, recombinant OTUB1 (1 μM), AMSH (1 μM), OTULIN (1 μM), Cezanne (200 and 500 nM), vOTU 183 (5 μM) and USP21 (1 μM) DUBs (produced in-house and kindly provided by David Komander) were added to the GST-UBA purified ubiquitylated proteins in the DUB reaction mix for 60 min at 37 °C. The reaction was stopped by adding 5× denaturing Western blot sample buffer.

**RT-PCR analysis**. WT and $Il1b^{K133R/K133R}$ BMDMs ($3 \times 10^6$) were untreated or treated with LPS (50 ng/ml) for 1–3 h. After the indicated time points, cells were harvested, washed with PBS and total RNA extracted (Isolate II RNA mini kit, Bioline). cDNA was synthesized from 1 μg of total RNA using Oligo (dT) and Super Script Reverse Transcriptase III (Bioline). GoTaq qPCR Master Mix (Thermofischer, 4367659) (5 μl) was added to cDNA (1 μl) and gene-specific forward and reverse primers (5 μM, 4 μl). cDNA was amplified for 50 cycles in a Viia7 real-time PCR machine. The expression of mRNA was normalized to endogenous HPRT expression and unstimulated cells. The primers used were; $Tnf$ (forward, 5′ ACTGAACTTCGGGGTGATCG 3′ and reverse, 5′ TGATCTGAGTGT-GAGGGTCTGG 3′), $Il1b$ (forward, 5′ GCTACCTGTGTCTTTCCCGT 3′ reverse, 5′ ATCTCGGAGCCTGTAGTGC 3′) and $Hprt$ (forward, 5′ TGAAGTACTCAT-TATAGTCAAGGGCA 3′ reverse, 5′ CTGGTGAAAAGGACC TCTCG 3′).

**Western blots**. Cell lysates and supernatants (reduced and denatured) were separated on 4–12% gradient gels (Invitrogen), proteins transferred onto nitrocellulose membrane (Amersham), and ponceau staining performed routinely to evaluate loading accuracy. Membranes were blocked with 5% skimmed milk in PBS containing 0.1% Tween 20 (PBST) for 1–2 h and then probed overnight at 4 °C with primary antibodies (all diluted 1:1000 unless noted otherwise): Mouse β-actin (Sigma; A-1978), pro and mature IL-1β (R&D Systems; AF-401-NA), pro-Caspase-1 and cleaved Caspase-1 (Adipogen; AG-20B-0042-C100), pro-Caspase-8 (in-house), cleaved Caspase-8 Asp387 (Cell Signaling; 9429), NLRP3 (Adipogen; AG-20B-0014-C100), ASC (Santa Cruz Biotechnology; Sc-22514-R), IL-18 (1:500 dilution, BioVision; 5180R-100), IL-1α (1:500 dilution, Cell Signaling; 50794), Ubiquitin (Cell Signaling; 3933), K48-linked polyubiquitin (Merck clone Apu 2; 05-1307), K63-linked polyubiquitin (Merck clone Apu 3; 05-1308), K11-linked polyubiquitin (MABS107-I, clone 2A3/2E6; Merck) and Mcl-1 (Cell Signaling; 5453). Relevant horseradish peroxidase-conjugated secondary antibodies were applied for at least 1–2 h. Membranes were washed 4–6 times in PBS-Tween between antibody incubations and antibodies were either diluted in PBS-Tween containing 5% skimmed milk. Membranes were developed using ECL (Millipore) on an X-OMAT developer (Kodak) or using the ChemiDoc Touch Imaging System (Bio-Rad) and Image Lab software.

**Reporting summary**. Further information on research design is available in the Nature Research Reporting Summary linked to this article.

## Data availability

Mass spectrometry proteomics data have been deposited in ProteomeXchange Consortium via the PRIDE partner repository under the accecssion code PXD021305. Extracted peak lists were searched against the reviewed *Homo sapiens* database (UniProt, March 2015; https://www.uniprot.org/proteomes/UP000005640) containing murine IL1β sequence (P10749). The structure of IL-1beta is shown in Fig. 4e (PDB: 2MIB at https://www.rcsb.org/structure/2mib). Source data are provided with this paper.

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

## Acknowledgements

We gratefully acknowledge grant support from the National Health and Medical Research Council (NHMRC) of Australia: project grants (1145788 to J.E.V., K.E.L. and J.M.M.; 1101405 to J.E.V.; 1162765 to K.E.L.), Ideas grants (1183070 to J.E.V.; 1181089 to K.E.L.) and fellowships (1172929 to J.M.M.; 1141466 to J.E.V.). K.E.L. is an Australian Research Council (ARC) Future Fellow (FT190100266). J.S.P. is supported by an Australian NHMRC Career Development Fellowship (1159230). MJH is and NHMRC Senior Research Fellow (1156095) and supported by the Leukemia and Lymphoma Society SCOR grant 7015–18. The generation of the *Il1b* $^{K133R/K133R}$ mice used in this study was supported by the Phenomics Australia (PA) and the Australian Government through the National Collaborative Research Infrastructure Strategy (NCRIS) program. This work was also supported by operational infrastructure grants through the Australian Government Independent Research Institute Infrastructure Support Scheme (9000653) and the Victorian State Government Operational Infrastructure Support, Australia.

## Author contributions

The project was conceived by J.E.V. and K.E.L., the experiments designed by S.L.V., R.F., J.E.V., K.E.L., A.K., and L.F.D., the manuscript written by J.E.V., K.E.L. The experiments were performed by S.L.V., R.F., K.E.L., J.E.V., M.R., D.F., Z.L., D.S., G.E., A.J.V., J.S.P., and L.F.D. Expert advice, essential mice and reagents were provided by M.H., J.M.M., and A.I.W. All authors assisted with data interpretation and manuscript editing.

## Competing interests

The authors declare no competing interests.
