## [Peer Review File · Nature Communications]

REVIEWER COMMENTS

Reviewer #1 (Remarks to the Author):

Post-translational modifications (PTMs) of inflammasome components have been emerging as one of the major subjects in the field of inflammasome studies. Ubiquitination is one of the PTMs that has been shown to regulate NLRP3, ASC, and other components of inflammasomes. In this study, Vijayaraj et al. first report that pro-IL-1 β is also regulated by ubiquitination during the priming phase. Specifically, the authors observed that IL-1 β undergoes K48-, K63-, and K11-linked polyubiquitination using TUBE assay, and identified IL-1 β K133 as the ubiquitination site. One of the outcomes of IL-1 β ubiquitination is to target it for proteasome-mediated degradation. The data presented are technically sound, and provide an additional evidence how ubiquitination controls the activation of inflammasomes.

Although the study is potentially interesting for the readers of Nature Communications, there are major and minor points which need to be addressed in the manuscript prior to publication.

Major points:

1. It is essential for the authors to identify potential E3 ubiquitin ligase(s) that ubiquitinates pro-IL-1 β . It is unclear why cIAP was selected to use in in vitro pro-IL-1 β ubiquitination assay.
2. Figure 2D: The authors should probe the membrane with anti-K11 ubiquitin antibody.
3. Is IL-1 β K133 binds to K48-, K63-, or K11-linked polyubiquitin chains? The authors should use WT and Il1bK133R BMDMs primed with LPS in the presence of MG-132, and then IP with anti-IL-1 β , and blot with anti-K48, K63, and K11 specific antibodies, respectively.
4. What are other potential ubiquitination sites? What are the biological relevance of K63- and K11-linked ubiquitination of IL-1 β if IL-1 β K133 binds to K48-linked polyubiquitin chains which targets IL-1 β to the proteasome for degradation.
5. Figure 3: It seems that pro-IL-1 β undergoes both caspase-1-mediated cleavage and proteasome-mediated degradation upon ATP and nigericin stimulation (Fig. 3A and B). If pro-IL-1 β undergoes ubiquitination upon ATP stimulation (Fig. 3C), then what is the potential E3 ubiquitin ligase(s)?
6. The authors should lyse the BMDMs primed with LPS, lyse them in RIPA buffer containing SDS under the denaturing condition to dissociate the contaminating proteins that are non-specifically bind to IL-1 β , and blot them with anti-ubiquitin, anti-K48, anti-K63, and anti-K11 antibodies, respectively, to confirm the data obtained from TUBE. The current data presented in Supplemental Figure 2 do not support whether IL-1 β K133 is the site that binds to K48 linked polyubiquitin chains, and whether multi-monoubiquitin chains bind to IL-1 β .
7. The authors should use LPS-induced endotoxemia model (survival rate and serum IL-1 β , TNF- α , IL-6) to verify the in vivo biological relevance of their findings on IL-1 β ubiquitination.

Minor points:

1. It has been shown that TRIM31 mediates NLRP3 ubiquitination and degradation during the priming phase, which is similar to pro-IL-1 β reported by the authors. However, NLRP3 degradation does not seem to occur in this report (Fig. 2A and B). It is unknown why?
2. It is unclear what the button of right panel of Figure 1A is. The actin expression in the lysate input is not shown.
3. How can the authors rule out the possibility that pro-IL-1 β may undergo multi-monoubiquitination (Fig. 3D).
4. Fig. 2C: Pro-IL-1 β is not increased upon MG-132 treatment (Fig. 2C, left panel). A light exposure may be helpful to clarify this concern. The authors should provide loading controls for cell lysate input.
5. Actin loading control should be provided for all lysate input.

Reviewer #2 (Remarks to the Author):

In this study the authors present evidence suggesting that the IL-1 β cytokine is targeted for ubiquitination and proteasomal degradation to limit inflammation. They show that proIL-1 β is ubiquitinated in response to inflammasome priming and activation signals and rapidly degraded by the proteasome. They identified K133 of proIL-1 β as a target site for ubiquitination in 293T cells after treatment with MG132 followed by mass spectrometry. To provide physiological relevance for their findings, they generated mice with K133R mutant proIL-1 β and found that these mice display increased proIL-1 β stability and bioactive IL-1 β production following inflammasome activation. Overall, although this study confirms two previous studies (refs 17 and 54) that showed ubiquitination of proIL-1 β on K133, unlike the previous studies this one demonstrates convincingly that preventing ubiquitination of K133 by mutating it to arginine does not impact processing of proIL-1 β by caspase-1 but increases its stability. As a consequence, this new study shows that there is more production of mature IL-1 β in mice carrying the K133R mutation in response to inflammasome stimulation. This new study is important and interesting because it clarifies the role of proIL-1 β ubiquitination in the inflammatory response and provides clear in vivo evidence on the impact of K133 ubiquitination on the stability of proIL-1 β and production of mature IL-1 β cytokine.

Major comments

1. Fig. 3D. ATP treatment appears to augment LPS-induced IL-1 β ubiquitination. Does nigericin treatment have the same effect? This question needs to be addressed because ubiquitination is driven by ATP. The exogenously added ATP could be responsible for enhancing the ubiquitination reaction in the cell independent of its effect on the NLRP3 inflammasome.
2. Fig. 3D. Does ATP or nigericin induce IL-1 β ubiquitination without LPS or PamCSK4 treatment. I understand that TLR stimulation is required for IL-1 β induction but this experiment can be done in immortalized BMDMs stably expressing IL-1 β or in neutrophils stimulated with IFN γ .
3. Although the K133 ubiquitination site was identified in 293T cells overexpressing IL-1 β , this site was not validated in BMDMs to confirm that it is indeed ubiquitinated under a more physiological setting in immune cells and in response to inflammasome stimulation.
4. Previous studies (refs 17 and 54) showed that K133R mutation reduces proIL-1 β processing. The authors should further discuss the reasons for the discrepancy between their results and the previous ones.
5. Does ubiquitination of pro-IL-1 β K133 inhibit its processing by caspase-1? It is interesting that preventing formation of the salt bridge between K133 and D129 by mutating them to alanines inhibits processing by caspase-1. It is thus possible that ubiquitination of K133 should mimic the effect of K133A mutation because it neutralizes the positive charge on the K residue.

We thank the reviewers for their constructive comments about our work. We have carefully revised our manuscript entitled “**The inflammasome-activated cytokine IL-1 β is targeted for ubiquitylation and proteasomal degradation to limit its inflammatory potential**” and have incorporated significant new data to address the reviewers’ concerns. We would like to note that, despite ongoing COVID-19-related restrictions on institute access and activities, including enforced limitations on animal breeding, these new experiments are extensive and wide-ranging. They include, (i) immunoprecipitations of endogenous IL-1 β protein complexes and mass-spectrometry to identify potential interacting ubiquitin E3 ligases, (ii) CRISPR gene targeting and biological analysis of identified potential E3 ligases, (iii) analysis of ubiquitin chain linkages tethered to wildtype IL-1 β and IL-1 β K133R at the endogenous level in primary cells, (iv) complementation studies of IL-1 β -deficient macrophages with IL-1 β constructs to examine its ubiquitination in the absence of inflammasome priming ligands and (v) biochemical experiments to define if ubiquitylated IL-1 β complexes are protected, or not, from inflammasome-associated caspase-1 processing. Importantly, as outlined below, these **new** data provide additional novel insights into the regulation of IL-1 β , which we believe further enhances the significance and importance of our findings.

In addition, we would like to highlight that our study represents one of the very few to document both the *in vitro* and *in vivo* relevance of a single inflammasome post-translational modification; the decoration of IL-1 β lysine 133 with ubiquitin. Our biochemical and genetic analysis, including the generation of a novel IL-1 β point-mutant mouse, has resulted in several novel discoveries. Specifically:

1. We document how precursor IL-1 β is a short-lived protein that is targeted for ubiquitylation and proteasomal degradation in response to pathogen-derived inflammasome priming ligands and also in response to inflammasome activating agents.
2. We show that ubiquitylation of IL-1 β lysine 133 (K133), and a K133:D129 electrostatic interaction, are important for IL-1 β proteasomal targeting and hence its activation and inflammatory potential.
3. We demonstrate that endogenous IL-1 β is decorated with K11-, K48- and K63-linked ubiquitin chains. Moreover, our **new** data indicate that endogenous ubiquitylated IL-1 β , and heterogeneous IL-1 β complexes comprising unmodified and ubiquitylated IL-1 β , are protected from caspase-1 cleavage. These data demonstrate that, contrary to recent high-profile studies, endogenous ubiquitylated IL-1 β is very unlikely to have a role in promoting bioactive IL-1 β release. Instead our findings support a model whereby ubiquitylated IL-1 β protein complexes are targeted to the proteasome for degradation to reduce the pool of available precursor IL-1 β , and at the same time this ubiquitylation event acts to prevent its processing by caspase-1.
4. We generate an IL-1 β K133 ubiquitylation-deficient mutant mouse (IL-1 β K133R) thereby revealing the importance of this modification at the endogenous level in primary macrophages and neutrophils and also its physiological relevance *in vivo* following intraperitoneal LPS injections.

We hope that the reviewers appreciate the biochemical and genetic rigor of our revised study, which when combined with the novel findings, we believe makes it suitable for publication in

Nature Communications. We have addressed all the reviewer comments, shown in *italics*, below:

Reviewer #1

Post-translational modifications (PTMs) of inflammasome components have been emerging as one of the major subjects in the field of inflammasome studies. Ubiquitination is one of the PTMs that has been shown to regulate NLRP3, ASC, and other components of inflammasomes. In this study, Vijayaraj et al. first report that pro-IL-1 β is also regulated by ubiquitination during the priming phase. Specifically, the authors observed that IL-1 β undergoes K48-, K63-, and K11-linked polyubiquitination using TUBE assay, and identified IL-1 β K133 as the ubiquitination site. One of the outcomes of IL-1 β ubiquitination is to target it for proteasome-mediated degradation. The data presented are technically sound, and provide an additional evidence how ubiquitination controls the activation of inflammasomes.

Although the study is potentially interesting for the readers of Nature Communications, there are major and minor points which need to be addressed in the manuscript prior to publication.

Author Response: We thank the reviewer for appreciating the technical rigor and the new findings that our study uncovers, and for agreeing that it is suitable for publication in *Nature Communications*.

Major points:

R#1: *1. It is essential for the authors to identify potential E3 ubiquitin ligase(s) that ubiquitinates pro-IL-1 β . It is unclear why cIAP was selected to use in in vitro pro-IL-1 β ubiquitination assay.*

Answer 1: This is an important point to clarify. We selected cIAP1 for the E3 ubiquitin ligase in the recombinant *in vitro* assay as it has previously been associated with the regulation of inflammasome protein complexes (Labbe K *et al.* *Immunity* 2012). However, *in vivo*, cIAP1 (and the related cIAP2 and XIAP) are unlikely to represent the ubiquitin E3 ligases that modify IL-1 β with ubiquitin chains as we have observed that cIAP1, cIAP2 and XIAP inhibition using IAP antagonist compounds (911 [Compound A]; targets cIAP1/2 and XIAP or 711 [Birinapant]; targets cIAP1/2) does not prevent IL-1 β ubiquitylation (**Response Figure 1, below**).

Response Figure 1 (NEW data). LPS-induced IL-1 β ubiquitylation occurs in the absence of cIAP1, cIAP2 and XIAP. BMDMs were pre-treated with IAP antagonists (911, 1 μ M [targets cIAP1/2 and XIAP] or 711, 1 μ M [targets cIAP1/2]) for 15 minutes (min) prior to LPS (50 ng/ml) stimulation for 6 hours. Subsequently, ubiquitylated proteins were purified by TUBE and analyzed by western blot, as indicated. Note, we previously showed that 911 activates IL-1 β (cl. IL-1 β ~ 17 kDa) as a consequence of XIAP targeting whereas 711 does not, while both 911 and 711 trigger cIAP1/2 degradation (Lawlor KE et al., Nature Communications, 2015; Lawlor KE*, Feltham R*, Yabal M* et al., Cell Reports 2017).

To attempt to identify the ubiquitin E3 ligase targeting IL-1 β we immuno-precipitated endogenous IL-1 β protein complexes, using an IL-1 β antibody, from LPS treated BMDMs and interrogated the complexes by Mass spectrometry (MS). This identified several significantly enriched ubiquitin E3 ligases that purified with anti-IL-1 β in WT LPS-primed macrophages relative to unprimed control cells (that lack IL-1 β expression) (**Response Table 1, below**).

Response Table 1: Ubiquitin E3 ligases enriched in IL-1 β purified complexes *

Gene Name	Accession Number	Log 2 Protein Ratio
IL-1 β	P10749	5.097372581
Arih1	Q9Z1K5	3.101094068
Dtx3l	Q3UIR3-2	1.125588232
Rnf213	F7A6H4	1.600062586
Trim21	Q3U7K7	1.148967802

*IL-1 β antibody was cross-linked to protein G beads (5 mM BS3) and incubated with cell lysates derived from 20 million BMDMs that had been stimulated with LPS for 6 hrs with MG132 added in the last 2 hr. Purified IL-1 β and interacting proteins were eluted in 0.5% SDS, 5 mM DTT (confirmed by immuno-blot) and were subsequently analyzed by Mass spectrometry. Candidate proteins with Log 2 protein ratios above 1 indicated significant enrichment and were considered possible IL-1 β interacting proteins.

To test if the ubiquitin E3 ligases identified (**Response Table 1, above**) targeted IL-1 β for ubiquitylation, or regulated IL-1 β secretion, we generated iBMDMs lacking these E3 ligases via CRISPR/Cas9 (85-95% efficiency; **Response Figure 2** & not shown). We then purified ubiquitylated IL-1 β by TUBE following LPS priming, and also measured IL-1 β release by ELISA after NLRP3 inflammasome activation by nigericin or the IAP antagonist 911 (Compound A). However, the deletion of these ubiquitin E3 ligases had no impact on IL-1 β ubiquitylation or its release (**Response Figure 2**). Similarly, deletion of UBE3A (**Response Figure 2**), previously implicated in IL-1 β ubiquitylation (Niebler M *et al.* PLoS Pathogens 2013), also had no impact on IL-1 β modification or release (**Response Figure 2**). Interactions between enzymes, such as ubiquitin E3 ligases, and their target proteins can be transient, making co-immunoprecipitation strategies difficult. We are now attempting to identify the relevant IL-1 β ubiquitin E3 ligase via an arrayed CRISPR/Cas9 E3 ubiquitin ligase screen which covers the 800+ single and multi-subunit E3 ligase encoding genes. However, this is beyond the scope of the current study, which already documents new IL-1 β regulatory mechanisms, both *in vitro* and *in vivo* via the generation of a novel IL-1 β point mutant mouse.

Response Figure 2 (NEW data). Ubiquitin E3 ligase deletion and impact on IL-1 β secretion and ubiquitylation. Immortalized BMDMs expressing Cas9 and gRNAs targeting MS-identified ubiquitin E3 ligases (2 independent gRNAs per gene, as indicated) were treated with LPS (50 ng/ml) and nigericin (20 μ M, 1 hr) or 911 (1 μ M, 6 hr) and IL-1 β release into the cell culture supernatant measured by ELISA (**Left panel**), or after 6.5 hr of LPS stimulation ubiquitylated proteins purified by TUBE and analyzed by Western blot (**Right panel**, Unt.; untreated, no LPS control, cells. Cas9; Cas9 parental cells containing no gRNAs).

R#1: 2. Figure 2D: The authors should probe the membrane with anti-K11 ubiquitin antibody.

Answer 2: This is an excellent suggestion we would have already performed had we had access to this antibody at the time of conducting these experiments. As noted below in **Answer 3**, we have now acquired a K11-ubiquitin chain specific antibody, and present **new**

data from denaturing IL-1 β immuno-purifications that illustrate the presence of K48-, K63- and K11-linked ubiquitin chains on endogenous IL-1 β . Unfortunately, in specific regard to Figure 2D, the membranes from these experiments, now dating back several years, no longer exist. However, Figure 2D documents the ability of ubiquitin chain-specific DUBs to cleave ubiquitin tethered to IL-1 β , including Cezanne, the DUB specific for K11-linked ubiquitin chains. It is important to note that the selectivity of these DUBs have been extensively validated and they are routinely used in ubiquitin chain linkage analysis. The results clearly demonstrate that IL-1 β ubiquitin conjugates can be cleaved by DUBs that only target K11-, K63- or K48-linked ubiquitin chains.

R#1: 3. Is IL-1 β K133 binds to K48-, K63-, or K11-linked polyubiquitin chains? The authors should use WT and *Il1b*^{K133R} BMDMs primed with LPS in the presence of MG-132, and then IP with anti-IL-1 β , and blot with anti-K48, K63, and K11 specific antibodies, respectively.

Answer 3: This is a good suggestion which will validate the DUB analysis presented in Figure 2D and provide insight into how IL-1 β ubiquitylation might be altered in *IL-1 β ^{K133R/K133R}* cells. We have now performed these experiments (denaturing IPs) using wildtype (WT), *IL-1 β ^{K133R/K133R}* and *IL-1 β ^{-/-}* (control) BMDMs treated with LPS in the presence of proteasomal inhibition. Results demonstrate that IL-1 β from WT and *IL-1 β ^{K133R/K133R}* macrophages is similarly conjugated with K48-, K63- and K11-linked ubiquitin chains (**Revised manuscript Figure 7A, reproduced below for convenience**). Because IL-1 β ^{K133R} levels are increased compared to WT IL-1 β , the relative amount of IL-1 β ^{K133R} ubiquitylation may in fact be moderately decreased, particularly K11-linked ubiquitylation. Regardless, this finding is consistent with our data (**Figure 6A and Supp. 7A**), showing that IL-1 β must be targeted for ubiquitylation on other residues, which also likely contribute to its proteasomal targeting.

Figure 7A of revised manuscript (NEW data). BMDMs derived from mice of the indicated genotypes were treated with LPS (100 ng/ml) for 3 hr followed by proteasome inhibition

(PS341, 10 μ M) for an additional 3 hr. Cells were lysed for 30 min in DISC lysis buffer containing SDS, and lysates subsequently diluted with DISC to a final concentration of 0.04% SDS. Biotin conjugated IL-1 β antibody was used to purify IL-1 β , and its modification analyzed using the indicated ubiquitin chain-specific antibodies. One of 2 independent experiments.

R#1: 4. *What are other potential ubiquitination sites? What are the biological relevance of K63- and K11-linked ubiquitination of IL-1 β if IL-1 β K133 binds to K48-linked polyubiquitin chains which targets IL-1 β to the proteasome for degradation.*

Answer 4: This is a very relevant point which we have now speculated on in the revised manuscript. In summary, our data supports a model whereby the K63- and K11-linked ubiquitin chains tethered to IL-1 β are also involved in targeting IL-1 β for proteasomal degradation. First, the ubiquitin-proteasome system is predominantly responsible for the degradation of polyubiquitinated short-lived proteins, and our manuscript identifies IL-1 β as one of these. Second, as noted in our manuscript “Typically, proteins marked with K48-linked ubiquitin chains are targeted to the proteasome, although K63- and K11-linkages have also been implicated...^{22, 23, 24.}” In fact, reference 24 demonstrates that K63-linked polyubiquitin chains can recruit K48-linked chains for the subsequent assembly of K48/K63-branched ubiquitin chains, which can act to target proteins for proteasomal degradation. Third, other than cell cycle regulation, K11-linked ubiquitin chains are predominantly associated with proteasomal targeting, including innate immune components (e.g. reviewed in Huizen M and Kikkert M. *Front. Cell Dev. Biol.*, 2020). Fourth, despite pan-selective TUBE purification of ubiquitylated proteins from significant quantities of cell supernatant containing bioactive IL-1 β , we have been unable to detect the ubiquitylation of activated IL-1 β (despite K133, and several other lysines, being located in this fragment; see Supp. Figure 7A). Fifth, our **new** data demonstrate that not only are the ubiquitylated pools of precursor IL-1 β resistant to cleavage by caspase-1, but that bioactive IL-1 β in cell lysates is not ubiquitylated (see **Answer 17, below, including new Figure 7B**). This shows that ubiquitylated IL-1 β complexes cannot be recognized by caspase-1 for processing into its bioactive fragment. Collectively, this evidence indicates that K48-, K63- and K11-linked ubiquitin chains decorate IL-1 β protein complexes to prevent caspase-1 cleavage and also target IL-1 β for proteasomal degradation to limit inflammation.

In regard to other IL-1 β ubiquitylation sites, IL-1 β contains 19 lysine residues which are potential targets for this modification. Of these, 17 lysine residues (including K133) are conserved between human and mouse. We have mutated all 17 conserved IL-1 β lysines and performed overexpression studies in 293T cells. Remarkably, we still observed significant ubiquitylation of this IL-1 β mutant, indicating ubiquitylation of non-conserved lysine residues, or of non-canonical ubiquitylation sites (e.g. serine, threonine, cysteine). A recent study using over-expressed IL-1 β in 293T cells suggested that IL-1 β K30, K133, K205, K209 and K247 are all targeted for ubiquitylation (Zhang, Liu et al. *Nature Communications* 2018, **9**(1): 4225). However, as our manuscript demonstrates, it is critical that functional validation studies are performed at the endogenous protein level. Hence, in future studies we hope to define, at the endogenous level, other IL-1 β modification sites and their function.

R#1: 5. *Figure 3: It seems that pro-IL-1 β undergoes both caspase-1-mediated cleavage and proteasome-mediated degradation upon ATP and nigericin stimulation (Fig. 3A and B). If pro-*

IL-1 β undergoes ubiquitination upon ATP stimulation(Fig. 3C), then what is the potential E3 ubiquitin ligase(s)?

Answer 5: Please see **Answer 1**. As stated above, this is currently unknown and deletion of the most promising candidates did not identify them as the E3 ligase. We have commenced an 800 E3 ligase arrayed CRISPR screen, however this is an extensive undertaking and we feel it is beyond the scope of the current manuscript.

R#1: 6. *The authors should lyse the BMDMs primed with LPS, lyse them in RIPA buffer containing SDS under the denaturing condition to dissociate the contaminating proteins that are non-specifically bind to IL-1 β , and blot them with anti-ubiquitin, anti-K48, anti- K63, and anti-K11 antibodies, respectively, to confirm the data obtained from TUBE. The current data presented in Supplemental Figure 2 do not support whether IL-1 β K133 is the site that binds to K48 linked polyubiquitin chains, and whether multi- monoubiquitin chains bind to IL-1 β .*

Answer 6: Please see **Answer 3** and **Answer 10**.

R#1: 7. *The authors should use LPS-induced endotoxemia model (survival rate and serum IL-1 β , TNF- α , IL-6) to verify the *in vivo* biological relevance of their findings on IL-1 β ubiquitination.*

Answer 7: In our manuscript we inject a sub-lethal dose of LPS *in vivo* and document significantly increased amounts of IL-1 β , but not TNF or IL-6, in *IL-1 β ^{K133R/K133R}* mice by ELISA when compared to littermate WT control animals (Figure 7B-D of manuscript). Moreover, we go above and beyond nearly all published *in vivo* studies by examining the relative levels of *in vivo* precursor IL-1 β and its cleaved bioactive fragment (independent experiments with a collective number of 14 WT and 15 *IL-1 β ^{K133R/K133R}* animals) (**Figure 7F and Supp. Figure 7B of the revised manuscript, reproduced below for convenience**). These *in vivo* data clearly demonstrate increased precursor and bioactive IL-1 β in *IL-1 β ^{K133R/K133R}* mice upon LPS injection, consistent with its stabilization. This is important as the ELISA method to measure IL-1 β levels detects both precursor and bioactive IL-1 β (Conos S et al., PNAS 2017) and therefore does not necessarily indicate if IL-1 β is activated.

Figure 7F and Supp. Figure 7B of revised manuscript. LPS-induced IL-1 β is increased in IL-1 β K133R mice *in vivo*. Littermate WT and *IL-1 β ^{K133R/K133R}* mice were injected intra-peritoneally

with 100 µg of LPS. After 2 hr, peritoneal fluid was harvested and analyzed by immunoblot IL-1β. Each lane represents an individual mouse.

In regard to the suggestion that we validate the bioactivity of IL-1β by measuring survival in the LPS-induced endotoxic shock model, this is complicated by a number of factors. The widely accepted high-dose LPS (54 mg/kg) septic shock model is mediated via pyroptotic cell death, rather than activation of the NLRP3 inflammasome and IL-1β (Kayagaki N *et al.*, Nature 2011). Moreover, protection from lethality to moderate LPS doses (15 mg/kg) requires simultaneous targeting of both IL-1β and IL-18 (Vanden Berghe, T. *et al.*, American Journal of Respiratory and Critical Care Medicine, 2014), so elevated levels of IL-1β are unlikely to significantly alter the disease course. Possibly a better model (i.e. more reproducibly reliant on IL-1β) for studying the clinical consequences of IL-1β ubiquitylation would be to cross *IL-1β^{K133R/K133R}* mice onto the mouse model of human Muckle-Wells Syndrome (MWS), in which the MWS *Nlrp3* mutation (A350V) drives a more prolonged autoinflammatory pathology significantly (albeit not fully) dependent on IL-1β (Brydges, SD *et al.*, Immunity, 2009). However, COVID-19-associated research restrictions, resulting from extensive lockdowns in Melbourne (Australia), has precluded the generation of new crosses and reduced our ability to increase breeding for *in vivo* studies. We estimate it would take at least another 9-12 months to generate the relevant crosses and pursuing these studies for the current manuscript are unlikely to alter our conclusions to any major extent. Our results, as highlighted above, already document significant *in vivo* differences in *IL-1β^{K133R/K133R}* animals upon endotoxin exposure. As such, under the current conditions, we feel that the extensive time and workload required for these studies is not warranted.

R#1: Minor points:

1. It has been shown that TRIM31 mediates NLRP3 ubiquitination and degradation during the priming phase, which is similar to pro-IL-1β reported by the authors. However, NLRP3 degradation does not seem to occur in this report (Fig. 2A and B). It is unknown why?

Answer 8: The manuscript by Hui Song and colleagues (Song H *et al.* Nature Communications 2016) examined the relationship between TRIM31 and NLRP3 turnover. However, this study differed to our experiments as they used thioglycollate-elicited peritoneal macrophages compared to our use of bone marrow-derived macrophages, and their LPS and CHX pulse chase experiments only revealed obvious NLRP3 degradation 8-12 hours after CHX addition, whereas we observed significant IL-1β degradation within 2-4 hours of CHX treatment. Consequently, we did not measure levels of IL-1β or NLRP3 beyond 6 hours of CHX treatment (please see Figures 2A, 2B and 6A of our manuscript). Moreover, Song and colleagues do not control for the possibility that prolonged LPS and CHX treatment may induce cell death, which can result in overall protein loss. Indeed, in their CHX chase study presented in Fig. 2f, the degradation of other inflammasome-associated proteins, ASC and caspase-1, is also evident at later time points. We have never detected the ubiquitylation of these components upon pan-selective ubiquitin TUBE purification (e.g. Figure 1A of our manuscript). As noted in our manuscript, we always incorporate the pan-caspase inhibitor QVD-OPh in our LPS and CHX studies, which will prevent LPS and CHX-induced cell death, particularly at later time points. In our optimization assays, LPS/CHX induced approximately 25% cell death at 6 hr and QVD-OPh prevented this. In our experiments we therefore mitigate the potential impact of caspase

activity and cell death on overall protein levels, as clearly demonstrated by our protein loading controls (ponceau staining or actin) and the consistent, unchanging, levels of caspase-1, NLRP3 and ASC up to 6 hours post CHX treatment (**Figures 2A, 2B, 6A of our manuscript**; reproduced many times over several years).

R#1: 2. It is unclear what the button of right panel of Figure 1A is. The actin expression in the lysate input is not shown.

Answer 9: We thank the reviewer for seeking clarification and prompting improved labelling in this figure. This is the ubiquitin blot. It is worth mentioning that relative to TUBE purified ubiquitylated protein enrichment (left panel), ubiquitin smearing is almost undetectable in the cell lysate input (right panel), which demonstrates how efficient the TUBEs are in enriching and purifying the ubiquitylated proteome. The band sitting slightly below 37 kDa (Fig. 1A right panel) is likely a non-specific band (also present in Fig. 1C), and we have thus labelled it as such in the revised manuscript (this also serves to mark input protein levels as adequately as actin blotting can – please also see **Answer 12**).

R#1: 3. How can the authors rule out the possibility that pro-IL-1 β may undergo multi-monoubiquitination (Fig. 3D).

Answer 10: In the submitted manuscript we demonstrate that the deubiquitinase (DUB), vOTU, removes all ubiquitin chains from IL-1 β and completely collapses all high molecular weight forms of ubiquitylated IL-1 β (**Figure 2D, relevant probing reproduced below for convenience**). Because vOTU does not cleave the proximal ubiquitin moiety (or M1-linked ubiquitin) (Manuela HK et al., Nature Protocols, 2015) this shows that IL-1 β is polyubiquitylated.

Figure 2D of revised manuscript. vOTU cleaves all ubiquitin chains tethered to IL-1 β .

R#1: 4. Fig. 2C: Pro-IL-1 β is not increased upon MG-132 treatment (Fig. 2C, left panel). A light exposure may be helpful to clarify this concern. The authors should provide loading controls for cell lysate input.

Answer 11: This is an astute observation. We have consistently found that MG132 treatment only moderately, at best, increases precursor IL-1 β levels in BMDMs at these time points, as this blot reflects (longer LPS/MG132 incubations in BMDMs become problematic due to the onset of cell death). We believe this is because at any one time only a minor pool of endogenous IL-1 β is ubiquitylated and is in the process of being targeted for proteasomal degradation (see **Answer 17, below, including new data to support this statement**), suggesting that proteasomal targeting of IL-1 β is triggered once a threshold level of expression is reached. Consistent with this idea, and as documented in Figure 5B, the levels of overexpressed IL-1 β in 293T cells is more readily amplified, and observed, upon prolonged MG132 treatment. Our **new** data (see **Answer 14, below**) also supports this hypothesis, as it shows that expression of IL-1 β alone in IL-1 β knockout macrophages, in the absence of TLR signalling, suffices to trigger IL-1 β ubiquitylation, and this event is not altered even upon TLR engagement. In regard to the loading control, we have included the ponceau staining loading input control for Figure 2C in the revised manuscript. However, the best loading control for this experiment is ubiquitin; as it demonstrates robust purification of the ubiquitylated proteome across all samples (please also see **Answer 12, below**).

R#1: *Actin loading control should be provided for all lysate input.*

Answer 12: While actin loading is traditionally used as a protein loading control, Ponceau staining of the membranes is, arguably, a more than adequate substitute as it allows the visualization of many proteins (both of low and high abundance) within the one sample, and is more frequently being used by laboratories. We would note that every membrane in our manuscript includes either an actin blot or ponceau staining loading control. The only instances where we use neither is for some of the TUBE assays (e.g. Figure 3D) where the more appropriate ubiquitin probing loading control is used; as the interpretation of these data requires the ability to assess the levels of ubiquitylated protein purification between samples (and actin does not purify by TUBE).

Reviewer #2 (Remarks to the Author):

In this study the authors present evidence suggesting that the IL-1 β cytokine is targeted for ubiquitination and proteasomal degradation to limit inflammation. They show that proIL-1 β is ubiquitinated in response to inflammasome priming and activation signals and rapidly degraded by the proteasome. They identified K133 of proIL-1 β as a target site for ubiquitination in 293T cells after treatment with MG132 followed by mass spectrometry. To provide physiological relevance for their findings, they generated mice with K133R mutant proIL-1 β and found that these mice display increased proIL-1 β stability and bioactive IL-1 β production following inflammasome activation. Overall, although this study confirms two previous studies (refs 17 and 54) that showed ubiquitination of proIL-1 β on K133, unlike the previous studies this one demonstrates convincingly that preventing ubiquitination of K133 by mutating it to arginine does not impact processing of proIL-1 β by caspase-1 but increases its stability. As a consequence, this new study shows that there is more production of mature IL-1 β in mice carrying the K133R mutation in response to inflammasome stimulation. This new study is important and interesting because it clarifies the role of proIL-1 β ubiquitination in the inflammatory response and provides clear in vivo evidence on the impact of K133 ubiquitination on the stability of proIL-1 β and production of mature IL-1 β cytokine.

Author Response: We thank the reviewer for appreciating the rigour of our experiments and the importance of our findings.

Major comments

R#2 1. Fig. 3D. ATP treatment appears to augment LPS-induced IL-1 β ubiquitination. Does nigericin treatment have the same effect? This question needs to be addressed because ubiquitination is driven by ATP. The exogenously added ATP could be responsible for enhancing the ubiquitination reaction in the cell independent of its effect on the NLRP3 inflammasome.

Answer 13: This is an interesting point that we had not previously considered. Our **new** data shows that both ATP and nigericin treatment can increase IL-1 β ubiquitylation in caspase-1 deficient macrophages (**Supp. Figure 3B, reproduced below for convenience**). It is worth noting that we can only measure ATP- and nigericin-induced ubiquitylation and proteasomal targeting of IL-1 β in NLRP3 or caspase-1 deficient cells, as in inflammasome competent cells IL-1 β is rapidly cleaved and released, and cells are killed via pyroptosis. It therefore remains to be determined whether this observation has any physiological relevance. However, it is tempting to speculate that under conditions where cells do not rapidly die via caspase-1 mediated pyroptosis upon inflammasome triggering, this pathway may play an important role in limiting IL-1 β activity.

B.

Supplemental Figure 3B of revised manuscript (NEW data). Caspase-1-deficient BMDMs were generated, LPS primed (100 ng/ml, 3 hr), and then treated with ATP (5 mM) or nigericin (10 μ M) for the indicated times. Ubiquitylated IL-1 β was subsequently analyzed by TUBE purification and immunoblotting. 1 of 2 independent experiments.

R#2 2. Fig. 3D. Does ATP or nigericin induce IL-1 β ubiquitination without LPS or PamCSK4 treatment. I understand that TLR stimulation is required for IL-1 β induction but this experiment can be done in immortalized BMDMs stably expressing IL-1 β or in neutrophils stimulated with IFN γ .

Answer 14: Prompted by this query we have generated new data that, interestingly, show that stable IL-1 β expression alone in IL-1 $\beta^{-/-}$ macrophages suffices to trigger IL-1 β ubiquitylation, and that TLR engagement has no impact on this pattern of ubiquitin modification (**Figure 1C, reproduced below for convenience**). This indicates that TLR stimulation simply serves to induce the expression of IL-1 β and has no other significant role (e.g. induction of ubiquitin E3 ligases, ROS etc) in its ubiquitylation and proteasomal targeting. We have further demonstrated that ATP and nigericin can increase IL-1 β ubiquitylation and that this occurs independent of the upstream inflammasome machinery (i.e. ATP and nigericin-induced increases in IL-1 β ubiquitylation is observed at comparable levels in NLRP3 and/or caspase-1-deficient macrophages; see Figure 3D and Supp. Fig. 3B). Considering ATP- and nigericin-induced degradation of precursor IL-1 β is also prevented by proteasomal inhibition, we suggest that the observed ATP/nigericin-mediated increase in IL-1 β ubiquitylation likely contribute to proteasomal targeting.

Figure 1C of revised manuscript (NEW data). IL-1 β -deficient BMDMs were immortalized (iBMDMs) and subsequently infected with a retroviral IL-1 β cDNA vector containing an IRES upstream of GFP. IL-1 β complemented cells were FACs sorted (GFP) and stable cell lines established. Parental IL-1 $\beta^{-/-}$ or IL-1 β complemented (IL-1 $\beta^{-/-}$ + IL-1 β) iBMDMs were treated

with LPS (100 ng/ml) for the indicated times and ubiquitylated IL-1 β and NLRP3 examined by TUBE purification and immunoblotting. One of 2 independent experiments.

R#2 3. *Although the K133 ubiquitination site was identified in 293T cells overexpressing IL-1 β , this site was not validated in BMDMs to confirm that it is indeed ubiquitinated under a more physiological setting in immune cells and in response to inflammasome stimulation.*

Answer 15: This is correct. However, the fact that IL-1 β K133R is stabilized when expressed in 293T cells, and is also stabilized at the endogenous level in *IL-1 β ^{K133R/K133R}* mouse macrophages and neutrophils, as well as *in vivo*, argues that the ubiquitylation of IL-1 β K133 occurs in physiologically relevant cell types and is functionally important. Furthermore, references 17 and 54 also identified IL-1 β K133 as being targeted for ubiquitylation, including in BMDMs (ref 17). However, despite the fact that all 3 studies define IL-1 β K133 as a ubiquitylation site, our functional results do not agree with theirs, perhaps because they only studied IL-1 β K133 mutation in the context of a 293T overexpression system, compared to our endogenous knockin mutant analyses (please also see **Answer 16**, below).

R#2 4. *Previous studies (refs 17 and 54) showed that K133R mutation reduces proIL-1 β processing. The authors should further discuss the reasons for the discrepancy between their results and the previous ones.*

Answer 16: This is an important point; we thank the reviewer for prompting further discussion. References 17 and 54 only analyzed IL-1 β K133R processing in the context of 293T cell overexpression systems, while we validated IL-1 β K133R stabilization in both 293T cells and via the generation of *IL-1 β ^{K133R/K133R}* mice. To be frank, we are not sure why references 17 and 54 observed moderately reduced IL-1 β K133R processing/release in 293T cells. However, in the single western blot shown in reference 17, there is reduced precursor IL-1 β K133R levels compared to WT IL-1 β , which may reflect decreased plasmid transfection efficiency and readily accounts for its moderately reduced processing/release. This does not appear to be the case in the western blot shown in reference 54 (Figure 6B), although here the authors do not demonstrate equivalent caspase-1 expression between WT IL-1 β and IL-1 β K133R samples, which may also explain variation in IL-1 β processing. We would note that all our experiments using the 293T cell overexpression systems (including data presented in the manuscript) contain several controls not incorporated in reference 17 and 54; dose titrations of transfected caspase-1 and blots to show caspase-1 levels between samples, blots to show levels of both cell lysate and cell supernatant IL-1 β (precursor and cleaved), and GFP blots to account for IL-1 β plasmid levels between samples (internal control to account for differences in IL-1 β mutant stability).

R#2 5. *Does ubiquitination of pro-IL-1 β K133 inhibit its processing by caspase-1? It is interesting that preventing formation of the salt bridge between K133 and D129 by mutating them to alanines inhibits processing by caspase-1. It is thus possible that ubiquitination of K133 should mimic the effect of K133A mutation because it neutralizes the positive charge on the K residue.*

Answer 17: This is a very interesting idea that we have tested at length with several experimental approaches. The non-denaturing elution of ubiquitylated species of IL-1 β from

TUBE purifications, or UBA-GST, were inefficient and limited our ability to test its cleavage by recombinant caspase-1. Alternatively, while we found that IL-1 β bound to the TUBE agarose beads could not be efficiently cleaved by recombinant caspase-1, we could not eliminate the possibility that the TUBE beads interfered with IL-1 β recognition by caspase-1. Further to this, we also performed size exclusion chromatography to specifically isolate higher molecular weight Ub-IL-1 β species for *in vitro* caspase-1 cleavage, however, these experiments were complicated by the fact that non-modified IL-1 β co-fractionated with ubiquitylated IL-1 β .

Therefore, we treated BMDMs with LPS followed by proteasome inhibition to induce and enrich endogenous ubiquitylated IL-1 β species, and lysed cells in a caspase assay buffer containing 1% Triton-X100 and either the proteasome inhibitor PS341 alone, or together with the DUB inhibitor PR619. We then spiked recombinant active caspase-1 into these cell lysates to cleave endogenous IL-1 β , and subsequently purified the ubiquitylated proteins using pan-selective TUBEs (i.e. bind all ubiquitin chain types). We then analyzed whether TUBE purification of ubiquitylated IL-1 β was diminished by caspase-1 pre-treatment, which would indicate that it was able to be processed by caspase-1, or whether ubiquitylated IL-1 β levels remained similar between untreated and caspase-1 treated samples, which would indicate that caspase-1 cannot recognize ubiquitylated IL-1 β . Remarkably, results showed that i) ubiquitylated IL-1 β purification was identical between untreated and caspase-1 treated samples, and ii) bioactive p17 IL-1 β ubiquitylation was not detected, despite extensive caspase-1 processing of input cell lysate precursor IL-1 β (**Figure 7B of revised manuscript, reproduced below for convenience**). Moreover, the fraction of unmodified IL-1 β that we consistently co-purified with ubiquitylated IL-1 β was also not able to be processed by caspase-1. This indicates that ubiquitylated IL-1 β complexes cannot undergo caspase-1 mediated processing. The substantial processing and thus depletion of cell lysate (input) precursor IL-1 β , further indicates that the majority of IL-1 β in a cell at any one time is not ubiquitylated. We therefore suggest that the K11-, K48- and K63-linked ubiquitin chains that decorate precursor IL-1 β are all likely involved in its proteasomal targeting, and do not promote caspase-1 mediated processing or consequent release (see **Answer 4**). In addition, as indicated by the reviewer, these data also provide evidence that IL-1 β K133 ubiquitylation may disrupt the IL-1 β K133:D129 salt bridge, as mutant IL-1 β D129A or K133A, like ubiquitylated IL-1 β , is inefficiently processed by caspase-1 (Figure 5C of manuscript), and we have noted this possibility in the revised manuscript.

Figure 7B of revised manuscript (NEW data). BMDMs were primed with LPS (100 ng/ml) for 3hr followed by a further 3hr with PS341 (10 μ M). Cells were subsequently lysed in caspase assay buffer containing 1% Triton-X100 and either PS341 (10 μ M), or PS341 and PR619 (10 μ M), as indicated. Recombinant caspase-1 (5 Units) was added and cell lysates incubated at 37 $^{\circ}$ C for 80 min. Subsequently, samples were diluted with DISC (containing protease inhibitors and 2 mM NEM) and ubiquitylated proteins purified by TUBE and analyzed by immunoblot. One of 2 independent experiments.

REVIEWERS' COMMENTS

Reviewer #2 (Remarks to the Author):

The authors have satisfactorily addressed all my comments.

Reviewer #3 (Remarks to the Author):

In the manuscript, Vijayaraj et al. report that pro-IL-1 β undergoes K48, K63 and K11 linked ubiquitination and identify IL-1 β K133 as the ubiquitination site. Although the ubiquitination of pro-IL-1 β has been reported by several groups, this study provides novel functions of ubiquitination of pro-IL-1 β in priming stage of inflammasome, and more physiological relevance of ubiquitination and degradation of pro-IL-1 β in inflammation with K133R mutant proIL-1 β mice in vivo. In the revised version of the manuscript, the authors have made substantial additions to the experimental data. The authors have substantively addressed each comments made by Reviewer #1. I have one minor suggestion that the loading control (such as actin) should be provided for lysate input in Fig.7A and Supplementary Fig.4.

Response to Reviewer Comments

Reviewer #2

The authors have satisfactorily addressed all my comments.

Author Response: Thank you.

Reviewer #3

In the manuscript, Vijayaraj et al. report that pro-IL-1 β undergoes K48, K63 and K11 linked ubiquitination and identify IL-1 β K133 as the ubiquitination site. Although the ubiquitination of pro-IL-1 β has been reported by several groups, this study provides novel functions of ubiquitination of pro-IL-1 β in priming stage of inflammasome, and more physiological relevance of ubiquitination degradation of pro-IL-1 β in inflammation with K133R mutant proIL-1 β mice in vivo. In the revised version of the manuscript, the authors have made substantial additions to the experimental data. The authors have substantively addressed each comments made by Reviewer #1. I have one minor suggestion that the loading control (such as actin) should be provided for lysate input in Fig.7A and Supplementary Fig.4.

Author Response: Thank you. We have provided loading controls for lysate inputs in Fig. 7A and Supplementary Fig. 4 in the revised manuscript.